# Estimating countries' additional carbon accountability for closing the mitigation gap based on past and future emissions

Thomas Hahn [1] ✉, Johannes Morfeldt [2], Robert Höglund[3], Mikael Karlsson [4] & Ingo Fetzer [1,5]

Quantifying fair national shares of the remaining global carbon budget has proven challenging. Here, we propose an indicator—additional carbon accountability—that quantifies countries' responsibility for mitigation and $CO_2$ removal in addition to achieving their own targets. Considering carbon debts since 1990 and future claims based on countries' emission pathways, the indicator uses an equal cumulative per capita emissions approach to allocate accountability for closing the mitigation gap among countries with a positive total excessive carbon claim. The carbon budget is exceeded by 576 Gigatonnes of fossil $CO_2$ when limiting warming below 1.5 °C (50% probability). Additional carbon accountability is highest for the United States and China, and highest per capita for the United Arab Emirates and Russia. Assumptions on carbon debts strongly impact the results for most countries. The ability to pay for this accountability is challenging for Iran, Kazakhstan and several BRICS+ members, in contrast to the G7 members.

Currently, 195 parties have committed in the Paris Agreement to "holding the increase in the global average temperature to well below 2 °C above pre-industrial levels and to pursue efforts to limit the temperature increase to 1.5 °C" (Article 2.1)[1]. Any such target can be translated into a budget for the maximum amount of carbon dioxide emissions that can be emitted over time, given a specified probability[2]. Due to historical emissions, the remaining global carbon budget compatible with not exceeding the 1.5 °C level with at least 50% probability is small, and respecting the budget requires rapidly declining emissions[2,3].

The climate action plans of the parties to the Paris Agreement—the Nationally Determined Contributions (NDCs)[4]—can be seen as national claims on the remaining global carbon budget. In addition to the NDCs, 148 countries, responsible for 88% of global emissions, have proposed some kind of long-term targets such as a net zero or climate-neutral commitment[5]. If countries meet their NDCs and their net-zero targets (NZTs), the temperature increase could be limited to 1.7–1.8 °C[6]. However, many countries still lack implementation plans to

achieve their pledges and considering only countries with credible pledges points towards a 2.4 °C temperature increase[7].

The Paris Agreement expresses the fairness principle of common but differentiated responsibilities and respective capabilities (CBDR-RC), which should be considered "in the light of different national circumstances" (Article 2.2)[1]. The principle has been part of international climate policy since the adoption of the United Nations Framework Convention on Climate Change (UNFCCC) in 1992[8]. However, there is no agreement on how to operationalise this principle, other than that developed country parties should continue to take the lead (Articles 4.4 and 9.3)[1].

Consequently, several approaches have been used for conceptualising and calculating fairness in relation to national carbon budgets[9–12]. These include *Equal annual per capita emissions* for countries, based on multiplying the annual global emissions, for a pathway in line with a certain global carbon budget, with the national share of the global population. Equal annual per capita allocations can also be extended to account for historical emissions by instead

[1]Stockholm Resilience Centre, Stockholm University, Stockholm, Sweden. [2]Physical Resource Theory, Department of Space, Earth and Environment, Chalmers University of Technology, Gothenburg, Sweden. [3]Marginal Carbon AB, Stockholm, Sweden. [4]Climate Change Leadership, Department of Earth Sciences, Uppsala University, Uppsala, Sweden. [5]Bolin Centre for Climate Research, Stockholm University, Stockholm, Sweden. ✉e-mail: thomas.hahn@su.se

focusing on cumulative emissions, labelled as *Equal cumulative per capita emissions*[9], where summed emissions starting from a historic year are equal per capita. This captures the CBDR principle but without the 'respective capabilities'. A common approach to operationalize this RC part of the principle is to use a country's Gross Domestic Product (GDP) per capita in relation to the global GDP per capita as a proxy for a fair allocation[13]. *Grandfathering* approaches are based on national emission reductions from present levels, so that the global remaining carbon budget is allocated based on current emission shares[10]. *Per capita convergence*, sometimes called *Contraction and convergence*[14], starts with grandfathering and converges over time into an equal annual per capita approach by the end of a specified period, while staying within the global carbon budget[11,13].

More than half (56%) of the NDCs consider fairness in relation to the country's past, current and future shares of global emissions, drawing on a variety of equal per capita principles or relating to trends applying different metrics[4]. Since the NDCs are developed and adopted independently by each party, they can be seen as bottom-up approaches to fairness[15]. Governments' perceptions of what is "fair enough" are manifested in negotiations and commitments like NDCs[16]. At the same time, the NDCs constitute roadmaps with a starting point in current emission levels. This grandfathering characteristic attributes legitimacy to the present unequal levels of emissions and risks perpetuating inequality[11]. Conversely, national equal annual and cumulative per capita allocations emphasise equity and could be seen as top-down approaches from a global budget[15] but lack reference to national sovereignty and an agreed process for updating national commitments, which are the core of the NDCs[17].

The carbon debt, and with it the responsibility for historical emissions, varies among countries[18]. Future carbon claims compared to carbon budgets compatible with the Paris Agreement have also been analysed for the major world regions based on moral claims by developing countries combined with cost-effective, ambitious pathways for developed countries[12,19]. Moreover, studies use the equal cumulative per capita emissions approach to share the responsibility for Carbon Dioxide Removal (CDR) among countries, assuming that countries with large cumulative emissions have the highest responsibility[20,21]. However, the combined responsibility for historical emissions and future emission claims made by individual countries has not yet been estimated. Such an indicator could be used to assess the need for additional measures, including raised domestic and international emission reduction ambitions, as well as CDR, and to allocate responsibility for those additional measures.

The aim of this study is to estimate the additional carbon accountability for countries, defined as each country's responsibility to mitigate or remove $CO_2$, in addition to achieving its NDC and NZT pledges, to stay within its share of a specific global carbon budget allocated based on equal cumulative per capita emissions and accounting for carbon debts based on the same method. This additional carbon accountability indicator operationalises the common but differentiated responsibilities principle in the light of different national circumstances and complements previous research by combining bottom-up estimations of country emission pathways, based on pledges made by governments, with a top-down equal cumulative per capita approach. The ambition is to build on the Paris Agreement, being the internationally agreed climate treaty, and improve the fairness of its implementation by acknowledging common but differentiated responsibilities for past and future emissions.

On this basis, the article presents calculations of the additional carbon accountability for 37 countries, for the years 1990–2070, based on a remaining global carbon budget for holding the global average temperature increase below 1.5 °C with 50% probability. We also analyse the countries' ability to pay the estimated costs of fulfilling this accountability, and evaluate the sensitivity of our method to (i) different starting years for historical responsibility, (ii) the chosen principle for allocating the remaining global carbon budget and for redistributing emission allowances not used by countries with ambitious climate targets, and (iii) a global carbon budget for holding the global average temperature increase below 2 °C with 83% probability. The results are thus relevant for the international climate negotiations and national climate policy design.

## Results
### Carbon debts and excessive carbon claims
First, we calculate both the carbon debt and the future excessive carbon claims for 37 countries (see definitions in Table 1 and results in Table 2). We classified the countries into four groups as defined by the income categories used by the World Bank (see Data sources). Note that the European Union (EU) is considered as one country. The 'Rest of the World' includes planned emissions by other countries (76%) and international transportation (24%) and is assumed to reach net zero emissions by 2070. Given a remaining global fossil carbon budget of 225 billion tonnes of carbon dioxide ($GtCO_2$) (see "Methods"), the global excessive carbon claim, exceeding the 1.5 °C carbon budget, is 576 $GtCO_2$, if all countries deliver upon their NDCs and NZTs.

All 15 high-income countries in our sample except Chile have a carbon debt, whereas the United States (139 $GtCO_2$) and the EU (50 $GtCO_2$) have the largest debts. Of the 13 upper-middle-income countries, only four—China, South Africa, Iran and Kazakhstan—have a carbon debt. The total carbon debt of these 18 indebted countries is 341 $GtCO_2$.

If we instead look to the future, the planned emission pathways 2023-2070 result in claims on the remaining carbon budget of 801 $GtCO_2$ (Supplementary Table 1). Figure 1a shows the notable differences in claims between China (30.8% of global claims), India (11.5%), the United States (7.8%), Russia (5.4%) and the EU (4.4%). However, carbon claims per capita are generally highest for high-income countries with some important exceptions including Iran and China (Fig. 1b and Table 3).

## Table 1 | Definitions of concepts

| Concept | Definition |
|---|---|
| Carbon debt | The difference between a country's actual 1990–2022 $CO_2$ emissions and its equal share in global emissions over that period is based on equal cumulative per capita emissions. |
| Excessive carbon claim | The difference between a country's cumulative, planned $CO_2$ emissions to achieve its targets—its carbon claim—and the country's allocated share in the global remaining carbon budget for 2023–2070 based on equal cumulative per capita emissions. |
| Total excessive carbon claim | The sum of a country's carbon debt and excessive carbon claims. |
| Additional carbon accountability | Each country's responsibility to mitigate or remove $CO_2$, in addition to its NDC and NZT pledges, is to stay within its share of a specific global carbon budget allocated based on equal cumulative per capita emissions and accounting for carbon debts based on the same method. Emission allowances not used by individual countries (i.e. countries with a negative total excessive carbon claim) are redistributed to other countries based on an equal cumulative per capita approach. |

The table lists definitions of the concepts of carbon debt, excessive carbon claim, total excessive carbon claim, and additional carbon accountability.

**Table 2 | Additional carbon accountability for achieving 1.5 °C with a 50% probability**

| Country | Carbon debt (MtCO$_2$) | Excessive carbon claim (MtCO$_2$) | Total excessive carbon claim (MtCO$_2$) | Additional carbon account-ability (MtCO$_2$) | Additional carbon account-ability per capita (tCO$_2$) |
|---|---|---|---|---|---|
| **High-income countries** | | | | | |
| Norway | 681 | 230 | 910 | 482 | 78 |
| Switzerland | 259 | 319 | 578 | 0 | 0 |
| Singapore | 874 | 707 | 1581 | 1151 | 185 |
| United States | 138,705 | 53,844 | 192,549 | 167,127 | 453 |
| Australia | 8974 | 4886 | 13,860 | 11,704 | 374 |
| Canada | 13,046 | 5678 | 18,724 | 15,638 | 349 |
| United Arab Emirates | 3848 | 5072 | 8920 | 8154 | 733 |
| New Zealand | 474 | 295 | 769 | 369 | 63 |
| United Kingdom | 7590 | 2436 | 10,027 | 5155 | 73 |
| European Union | 50,351 | 25,780 | 76,131 | 47,563 | 115 |
| Japan | 21,328 | 11,698 | 33,025 | 25,698 | 242 |
| South Korea | 9510 | 7392 | 16,902 | 13,740 | 299 |
| Saudi Arabia | 10,735 | 13,738 | 24,473 | 21,303 | 463 |
| Russia | 33,726 | 39,835 | 73,560 | 64,327 | 480 |
| Chile | −319 | 880 | 561 | 0 | 0 |
| **Upper-middle income countries** | | | | | |
| Argentina | −596 | 2492 | 1896 | 0 | 0 |
| Costa Rica | −429 | −10 | −439 | 0 | 0 |
| China | 25,208 | 215,184 | 240,392 | 150,379 | 115 |
| Kazakhstan | 4767 | 5136 | 9903 | 8200 | 332 |
| Mexico | −1,666 | 11,171 | 9505 | 0 | 0 |
| Türkiye | −855 | 7804 | 6949 | 535 | 6 |
| Brazil | −14,973 | 2199 | −12,775 | 0 | 0 |
| Peru | −2978 | 95 | −2883 | 0 | 0 |
| Thailand | −2790 | 5294 | 2504 | 0 | 0 |
| South Africa | 6132 | 4551 | 10,682 | 5796 | 82 |
| Colombia | −3,909 | 151 | −3759 | 0 | 0 |
| Indonesia | −21,376 | 16,196 | −5180 | 0 | 0 |
| Iran | 4823 | 30,003 | 34,826 | 28,210 | 294 |
| **Lower-middle income countries** | | | | | |
| Egypt | −6546 | 5668 | −878 | 0 | 0 |
| Viet Nam | −8131 | 4796 | −3335 | 0 | 0 |
| Philippines | −10,254 | 1282 | −8972 | 0 | 0 |
| Morocco | −3098 | 1194 | −1904 | 0 | 0 |
| India | −123,182 | 53,337 | −69,846 | 0 | 0 |
| Nigeria | −18,970 | −5444 | −24,415 | 0 | 0 |
| Kenya | −5191 | −1242 | −6433 | 0 | 0 |
| **Low-income countries** | | | | | |
| Ethiopia | −11,919 | −4532 | −16,451 | 0 | 0 |
| Gambia | −252 | −91 | −342 | 0 | 0 |
| Rest of world | −103,596 | 47,512 | −56,084 | 0 | 0 |
| World | 0 | 575,534 | 575,534 | 575,534 | 61 |

The table shows results for the analysed countries, ordered according to falling GDP per capita, using the World Bank categories (see Data sources). The concepts are defined in Table 1. The additional carbon accountability per capita is based on the country's average population during 2023–2070.

The large total excessive carbon claims (Table 1) by high-income countries together originate mainly from carbon debts, while middle-income countries often have larger excessive carbon claims 2023–2070 (Fig. 2). Only five countries—Nigeria, Ethiopia, Kenya,

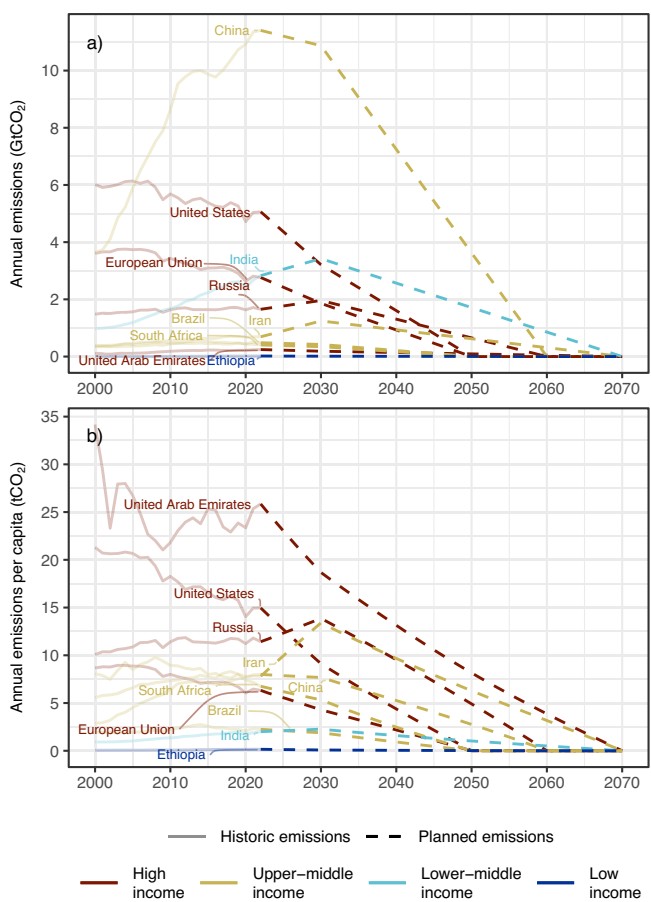

**Fig. 1 | Historical and planned emissions for a selection of different income countries. a** Annual emissions for selected countries. **b** Annual emissions per capita for selected countries. The graphs show historical emissions during 2000–2022 (solid lines) and planned emissions assuming that the countries' NDCs and NZTs are achieved (dashed lines). The countries have been selected to represent all income levels in the analysis, as well as different country sizes. Data for all analysed countries are provided in the online repository and assumptions for planned emissions are provided in Supplementary Table 2.

Gambia, and Costa Rica—have no excessive carbon claims 2023–2070. These countries also do not have a carbon debt.

### Estimating countries' additional carbon accountability

The additional carbon accountability is an indicator of holding a country accountable for its excessive historical and future carbon claims, on top of achieving its national targets. The sum of only positive total excessive carbon claims is 789 GtCO$_2$ but the global net sum of excessive carbon claims is 576 GtCO$_2$, since 14 countries plan to emit less than what follows from an equal cumulative per capita emission approach for 1990–2070. The difference, referred to as the emission allowances pool, corresponds to 213 GtCO$_2$ or 27% of the large emitters' total excessive carbon claims. The emission allowances pool is redistributed to countries with positive total excessive carbon claims based on equal cumulative per capita access to the pool for countries' future populations.

United States (167 GtCO$_2$) and China (150 GtCO$_2$) have by far the largest additional carbon accountability (Table 2). In total, 18 of the 37 countries in our sample are accountable for increasing their ambitions to stay within their equal share of the global carbon budget for 1.5 °C. The 19 countries with no additional accountability include all low-income and lower-middle-income countries, two high-income countries (Switzerland and Chile), and eight upper-middle-income countries: Argentina, Costa Rica, Mexico, Brazil, Peru, Thailand, Colombia, and Indonesia (Table 2).

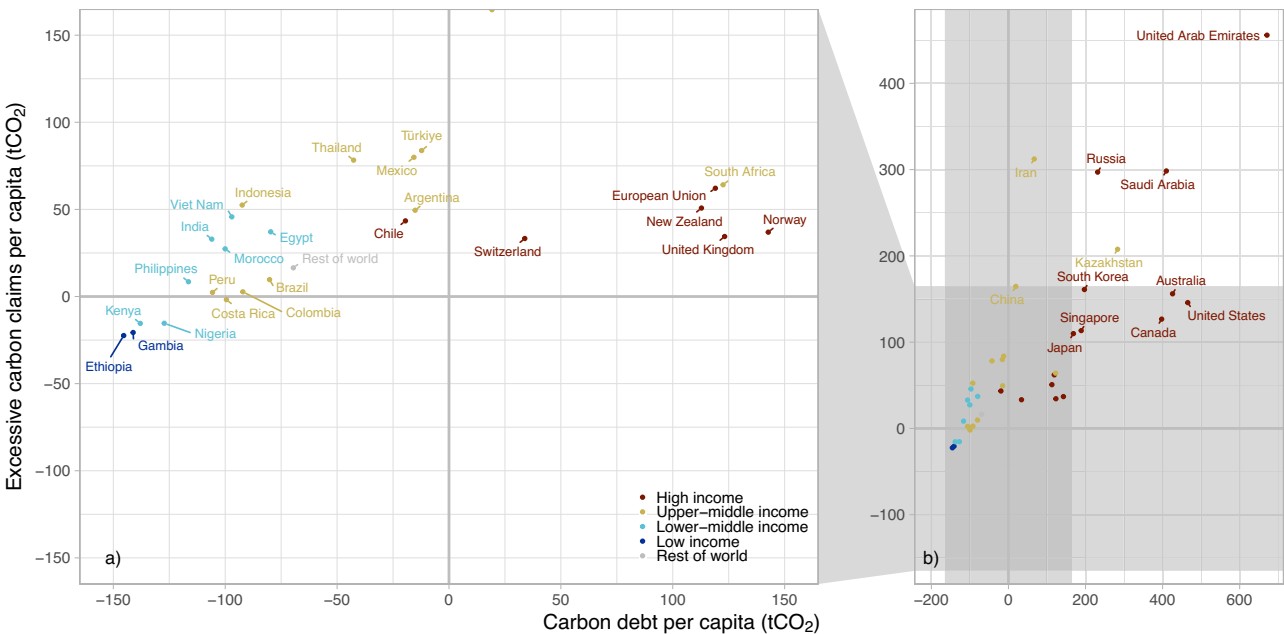

**Fig. 2 | Excessive carbon claims and carbon debt per capita.** The figure shows the excessive carbon claims per capita during 2023–2070 (y-axis) and carbon debt per capita during 1990–2022 (x-axis) for countries of different income levels (colour), based on average populations during the analysed periods. The left panel **a** is an expansion of the right panel (**b**). Countries in the upper right quadrant have a positive carbon debt (i.e. higher emissions during the period than the cumulative per-capita allocation of the global cumulative emissions over the same period) and positive excessive carbon claims (i.e. their planned emissions exceed the cumulative per-capita allocation of the remaining carbon budget 2023–2070). A similar figure for country-level excessive carbon claims and carbon debts is provided in Supplementary Fig. 1 and equivalent figures for a temperature target of 2 °C at 83% probability are provided in Supplementary Figs. 2 (per capita) and 3 (country-level).

The additional carbon accountability per capita may be a more adequate measurement, which puts the United Arab Emirates at the top, followed by Russia, Saudi Arabia, the United States, Australia, Canada, and Kazakhstan—all of them major fossil fuel producers.

For high-income countries, the additional carbon accountability for the 1.5 °C budget (Table 2) is typically larger than their planned future emissions (Supplementary Table 1). It means that these countries cannot meet their accountabilities with more ambitious domestic emission reductions alone but would need to also increase ambitions for CDR or for emission reductions in other countries.

Five of 18 countries with additional carbon accountability have larger planned future emissions than additional accountabilities: Türkiye, China, New Zealand, Iran and South Africa. They could theoretically fulfil their entire additional carbon accountability with stricter domestic emission reductions than stipulated in their NDCs and NZTs. For China, New Zealand, Iran and South Africa, this would however require a large portion of their planned future emissions, hence CDR is probably needed too.

Our results also show that even if all countries in our dataset with additional carbon accountability would reduce their emissions to zero today, the global carbon budget for 1.5 °C would still be exceeded. Future emissions from countries without additional accountability and in the Rest of the World category are large enough for the 1.5 °C budget to be exceeded.

**Analysis of results for G7 and BRICS+ countries**

In geopolitical terms, the G7 countries (United States, European Union, United Kingdom, Canada and Japan) together have a carbon debt (231 GtCO₂) more than double their excessive carbon claims 2023–2070 (99 GtCO₂). Conversely, the nine BRICS+ countries (Brazil, Russia, India, China and South Africa, as well as the four confirmed new members United Arab Emirates, Iran, Egypt and Ethiopia) have a notable negative carbon debt (−83 GtCO₂), mainly due to India, but large excessive carbon claims (351 GtCO₂), mainly due to China (Table 3).

Our analysis suggests that the statement above, that the grandfathering characteristic of NDCs and NZTs attributes legitimacy to the present unequal levels of emissions and risks perpetuating inequality[11], is only partially true. The timing of countries' NZTs plays a large role in future cumulative emissions. For example, if we assume that a country chooses 2070 as NZT over 2050 and that their emissions pathway is linear from 2023 onwards towards the target, then choosing net zero by 2070 implies 71% higher cumulative emissions compared to choosing 2050. Many large, developed countries take the lead in mitigation, hence respecting Article 4.4 of the Paris Agreement. Indeed, we found that two middle-income countries in the BRICS+ group—Iran and China—have larger carbon claims (planned emissions) per capita than any of the G7 countries (Table 3).

**Robustness of the additional carbon accountability**

We tested how sensitive our indicator is to different assumptions by varying: (i) the extent of historical responsibility (i.e. starting year for estimating the carbon debt), (ii) the allocation principles (i.e. the principle used to estimate a country's fair share in the remaining carbon budget and for redistributing the remaining emission allowances pool), and (iii) the global carbon budgets (i.e. implying a different long-term temperature target with a different probability). The methods used for the alternative allocation principles are based on previous studies[10,22] and described in Supplementary Note 1.

First, similar to the Climate Equity Reference Framework[23], using historical emissions for countries' responsibilities, we show that an earlier starting year for the carbon debt calculations generally implies larger additional carbon accountability for high-income countries and lower additional carbon accountability for middle-income countries, and vice versa for later starting years (Supplementary Fig. 4). Supplementary Fig. 4 shows the additional carbon accountability, additional carbon accountability per capita, and the changes compared to the main case for starting years 1960, 1980, 1990, and 2000, and a case with no carbon debt. The exceptions to this pattern are New Zealand,

**Table 3 | Differentiated responsibilities for past and future emissions among G7 and BRICS+ countries**

| Countries | Share of world population 2023–2070 | Carbon debt 1990–2022 (MtCO$_2$) | Excessive carbon claim 2023–2070 (MtCO$_2$) | Total excessive carbon claims (MtCO$_2$) | Planned emissions 2023–2070 per capita (tCO$_2$) |
|---|---|---|---|---|---|
| United States | 3.92% | 138,705 | 53,844 | 192,549 | 170 |
| European Union | 4.41% | 50,351 | 25,780 | 76,131 | 86 |
| United Kingdom | 0.75% | 7590 | 2436 | 10,027 | 58 |
| Canada | 0.48% | 13,046 | 5678 | 18,724 | 151 |
| Japan | 1.13% | 21,328 | 11,698 | 33,025 | 134 |
| Sum G7 | 10.69% | 231,020 | 99,436 | 330,456 | |
| Brazil | 2.41% | −14,973 | 2199 | −12,775 | 34 |
| Russia | 1.43% | 33,726 | 39,835 | 73,560 | 321 |
| India | 17.23% | −123,182 | 53,337 | −69,846 | 57 |
| China | 13.89% | 25,208 | 215,184 | 240,392 | 189 |
| South Africa | 0.75% | 6132 | 4551 | 10,682 | 88 |
| United Arab Emirates | 0.12% | 3848 | 5072 | 8920 | 480 |
| Iran | 1.02% | 4823 | 30,003 | 34,826 | 336 |
| Egypt | 1.62% | −6,546 | 5668 | −878 | 61 |
| Ethiopia | 2.15% | −11,919 | −4532 | −16,451 | 2 |
| Sum BRICS+ | 40.62% | −82,884 | 351,316 | 268,432 | |

The table shows results for the countries included in G7 and BRICS+. The concepts are defined in Table 1. The planned emissions per capita is based on the country's average population during 2023–2070.

South Korea, and Kazakhstan which show an opposite trend. For some countries, the choice of starting year for historical responsibility only makes a small difference (Norway, Singapore, United Arab Emirates, Saudi Arabia, and South Africa).

To give a few examples of the results for individual countries, the starting year of 1980 resulted in a 24% increase in the additional carbon accountability for the European Union and a 17% increase for the United States, but a 38% decrease for China. Going as far back as 1960, results in an increase of 65%, an increase of 49%, and a decrease of 97%, for the European Union, the United States and China, respectively. Including no carbon debt at all has the opposite effect, where high-income countries are assigned notably lower additional carbon accountability. In this case, the additional carbon accountability for the European Union and the United States are 47% and 68% lower than in the main case, respectively, but 42% higher for China. Shifting from using 1960 for historical responsibility, to ignoring it, implies 195 GtCO$_2$ lower additional carbon accountability for the United States and 209 GtCO$_2$ higher for China, hence assumptions about carbon debts have significant effects.

Second, we analyse the impact of different aspects of the chosen allocation principle—the principle used for allocating the carbon budget (Supplementary Fig. 5), and the principle used for redistributing the emissions allowances pool (Supplementary Fig. 6). Note that these figures assume all other parameters to be equal to the main case. The results show that the impact on the results of choosing a different allocation principle for the remaining carbon budget (i.e. capability, contraction and convergence, grandfathering, and equal cumulative per capita) is relatively small, with opposite patterns for the EU and the USA. Generally, the impact of the choice of allocation principle for the carbon budget is low (up to 12% change) for countries with additional carbon accountability over 100 tCO$_2$ per capita in the main case (Supplementary Fig. 5). The impact is larger for the choice of principle for redistributing the emission allowances pool (up to 41% change) for countries with additional carbon accountability over 100 tCO$_2$ per capita in the main case (Supplementary Fig. 6). For countries with low additional carbon accountability in the main case, the relative impact of allocation principle can be notably higher (especially for Türkiye that has a very low additional carbon accountability in the main case, but also New Zealand, United Kingdom, Norway, and South

Africa stand out). Both for the allocation of the remaining carbon budget and for the redistribution of the emission allowances pool, there are fewer clear patterns depending on income group as compared to the sensitivity analysis of the carbon debt. Hence, the results suggest that our indicator is generally less sensitive to the choice of allocation principle than the choice of considering countries' carbon debt.

Third, we show that the additional carbon accountability is notably reduced for all countries if a different temperature target−2.0 °C with 83% probability−is chosen, which implies a global carbon budget of 775 GtCO$_2$ (see "Methods"). This budget is more than three times as large as the 1.5 °C budget and only 26 GtCO$_2$ smaller than future planned emissions. Hence, national targets largely determine the results within the 2.0 °C budget and the equal cumulative per capita approach, which is decisive for allocating accountability within the 1.5 °C budget, allocates only small additional carbon accountability. For example, the total excessive carbon claim by the United States for the 2.0 °C budget is 171 GtCO$_2$ but its additional accountability is only 12 GtCO$_2$ (Supplementary Table 1). Note though that carbon budgets are associated with large uncertainties[24].

### Accountability and countries' ability to pay

If we return to the main results, we can analyse different countries' ability to pay to meet their additional carbon accountability. Mitigation or removal of the total excessive carbon claims of 576 Gt (1.5 °C budget) would cost 86,400 bn US Dollars (USD) if we assume an average cost of 150 USD/ton (see "Methods"). This would be 1728 bn USD per year on average over a 50-year period, which is 1.71% of the global GDP in 2022. This annual cost of returning below the 1.5 °C budget by 2100 may be lower than the annual global military expenditures which were 2240 bn USD in 2022[25].

The countries in our sample have different abilities to pay for their accountability. With high historical and future emissions and a low per capita income, Iran, followed by Kazakhstan and Russia, turned out to have the greatest economic challenges to meeting its additional carbon accountability (Table 4).

Five of the top seven countries in this table belong to the BRICS+ group. All high-income countries except Russia, Saudi Arabia and the United Arab Emirates have committed to net zero by 2050. Eleven

**Table 4 | The capability to take accountability**

| Countries | Cost of accountability as a share of 2021 GDP | Year of net-zero target (NZT) | GDP per capita (2021) | Cost of accountability per year as a percentage of GDP (2021) | Military expenditures per year as a percentage of GDP (2021) |
|---|---|---|---|---|---|
| Iran | 1178% | 2070* | 4084 | 39.28% | 1.5% |
| Kazakhstan | 624% | 2060 | 10,271 | 15.60% | 0.8% |
| Russia | 523% | 2060 | 12,522 | 13.09% | 3.6% |
| Saudi Arabia | 366% | 2060 | 24,316 | 9.14% | 7.2% |
| United Arab Emirates | 295% | 2070* | 44,332 | 9.82% | 5.5% |
| South Africa | 207% | 2050 | 7074 | 4.14% | 0.8% |
| China | 127% | 2060 | 12,618 | 3.16% | 1.6% |
| Canada | 117% | 2050 | 52,497 | 2.34% | 1.3% |
| South Korea | 113% | 2050 | 35,126 | 2.27% | 2.8% |
| Australia | 113% | 2050 | 60,697 | 2.25% | 2.1% |
| United States | 106% | 2050 | 71,056 | 2.13% | 3.4% |
| Japan | 77% | 2050 | 40,059 | 1.53% | 1.0% |
| European Union | 41% | 2050 | 38,721 | 0.82% | 1.3% |
| Singapore | 40% | 2050 | 79,601 | 0.80% | 2.5% |
| United Kingdom | 25% | 2050 | 46,870 | 0.49% | 2.1% |
| New Zealand | 22% | 2050 | 49,624 | 0.44% | 1.1% |
| Norway | 14% | 2050 | 93,073 | 0.29% | 1.7% |
| Türkiye | 10% | 2050 | 9743 | 0.20% | 1.9% |

The table shows the additional carbon accountability per capita for the 1.5 °C budget, and the costs related to increased mitigation or carbon dioxide removal efforts in line with the accountability. The latter is shown relative to the country's GDP, in total and per year (until 2100, i.e. beyond the country's NZT), and compared to military expenditures (in current USD 2021). Source for military expenditures is Stockholm International Peace Research Institute[25] for all countries except the European Union, which is sourced from Eurostat[44].
*Indicate that the net zero target is missing and 2070 is assumed for the country.

countries have an additional carbon accountability larger than 100% of their GDP. Between the year of net zero and 2100, the annual cost as a share of GDP for meeting the additional carbon accountability would be higher than the current military spending for ten countries. For these countries, the ability to finance their additional accountability by the year 2100 may be an economic challenge, although co-benefits from mitigation should also be considered[26].

## Discussion

Today, much of the fairness debate in international climate negotiations focuses on the multilateral responsibility of high-income countries to finance mitigation, and adaptation, as well as loss and damages in developing countries[4]. The results have been meagre so far. For example, the multilateral Green Climate Fund – supporting developing countries in raising and realising their NDCs and achieving low-emission pathways—has only managed to raise about 25 bn USD in total until December 2023, including pledges to pay during the period 2024–2027[27]. Moreover, the commitment by developed countries in 2009, at the 15th Conference of the Parties (COP15) of the UNFCCC, to mobilise 100 bn USD per year by 2020, is still not achieved. This collective goal is not well-defined, so for mitigation, we argue that a differentiated national responsibility, based on the additional carbon accountability, would be a both distinct and more fair principle for allocating responsibility to close the mitigation gap, on top of the NDCs and net-zero targets.

The ambition of this article is to help enhance climate fairness in the real world. The calculations assume that countries are accountable to the mitigation targets they have voluntarily committed to within the frames of the Paris Agreement, regardless of whether these commitments are more ambitious than what a particular theoretical allocation principle would suggest is fair. This is of course, like all assumptions of fair allocations of carbon budgets, a normative assumption[28]. Another normative point of departure is the suggestion that the countries with additional carbon accountability are collectively responsible for meeting the global target. Such a two-step approach to fairness is not

unique. For example, Holz et al.[23] have suggested "dual obligations" for countries by first meeting their own national targets and then allocating additional obligations to countries with excessive emissions (large cumulative emissions) and high capacity (economic wherewithal).

For the 1.5 °C carbon budget of 225 GtCO$_2$, our indicator clarifies the diverse accountabilities of different countries for their excessive carbon claims (576 GtCO$_2$). For high-income countries, especially G7 countries (Table 3), the accountabilities largely depend on countries' carbon debts, while future planned emissions are a more important factor for other countries. The equal share of the 1.5 °C carbon budget is negative for 17 countries (14 high-income plus Kazakhstan, South Africa and Iran) (see detailed calculations in the data repository), if their carbon debt for 1990–2021 is deducted. Such negative carbon budgets are however not an argument for abandoning national allocations of the carbon budget. On the contrary, it reveals that every ton of future emissions must be accompanied by at least one ton of removal in the near future, or additional reductions financed in countries with remaining budgets, which in turn provides strong incentives for faster mitigation.

Our assumption of including carbon debts based on historical emissions in the additional carbon accountability results in higher responsibility for most high-income countries. In contrast, the additional carbon accountability is only sensitive to the choice of allocation principle for the remaining carbon budget, such as grandfathering, contraction and convergence, and capacity (GDP/capita), for countries with relatively low Additional Carbon Accountability in the main case. Meanwhile, our sensitivity analysis suggests that the choice of principle for redistributing unused emission allowances between countries may have larger impacts for some countries.

### Climate finance and accountability

Financing enhanced mitigation in another country offers an alternative strategy to deliver action in line with countries' additional accountabilities. However, this must be additional to both countries' (the

**Table 5 | Global carbon budgets**

| Temperature limit, [°C] | Probability | Global carbon budget [GtCO$_2$] from 2023 | Global fossil carbon budget [GtCO$_2$] from 2023 |
|---|---|---|---|
| 1.5 | 50% | 250 | 225 |
| 1.7 | 67% | 500 | 475 |
| 2.0 | 83% | 800 | 775 |

The global carbon budgets are from January 2023 and adapted from Forster et al.[3] Note that the budget assumes reductions in other greenhouse gases between 2020–2050, such as CH$_4$ (50 %) and N$_2$O (25 %).

paying and the receiving) domestic mitigation plans, i.e. NDCs and NZTs. There may be compelling cost-efficiency and fairness arguments for high-emitting or high-income countries to help developing countries achieve their NDCs and NZTs (Articles 6 and 9 of the Paris Agreement) but this is not analysed specifically in this article.

Discussions of countries' capabilities often focus on countries' lack of capacity to pay for mitigation measures[9]. We acknowledge the need to "provide financial resources to assist developing country Parties" (Article 9) and such assistance may be needed also for some middle-income countries, which have considerable additional carbon accountability for the 1.5 °C budget on top of the challenges to achieve their national targets (Table 4). Our results suggest that embracing their additional accountability is economically challenging for several BRICS+ countries. The geopolitical implications can hardly be underestimated and deserve further studies.

The global costs for adaptation, losses and damages are most probably lower for 1.5 °C warming than for 2.0 °C[29]. It would be fair if the countries with reduced accountability within the 2.0 °C budget, compared to the 1.5 °C budget, would be held accountable for the increased costs of adaptations, losses and damages. Therefore, the 2.0 °C budget is not necessarily desirable for the high emitters, if fairness would guide accountability also for these costs.

## Methods
To achieve the aim of this study—estimating countries' additional carbon accountability—we proceed in three steps: calculating the country's carbon debt based on its historical emissions, estimating the country's excessive carbon claims on the remaining global carbon budget, and estimating the country's accountability to mitigate or remove CO$_2$, in addition to its NDC and NZT pledges. Each step is described below.

Some studies[30,31] advocate for using consumption-based emission accounting as a basis for fair allocations of the remaining carbon budget. Our analysis uses territorial emission accounting since the Paris Agreement is designed around national policies focused on emissions and removals within each country and consumption-based emission accounting is not easily incorporated into the Paris Agreement's transparency framework[32].

### National carbon debt
We estimate the national carbon debt by subtracting a country's equal cumulative per capita share of historical global emissions from its cumulative emissions over the period 1990–2022, as Eq. (1). This method uses a country's cumulative population ratio to the global cumulative population as a factor for estimating each country's historical emission allocation given a certain level of cumulative global emissions. The approach encompasses the population dynamics of each country over the period analysed, rather than relying on isolated yearly data points.

$$\text{Carbon debt} = \sum_{t=1990}^{2022} E_{\text{country}}(t) - \sum_{t=1990}^{2022} E_{\text{global}}(t) \cdot \frac{\sum_{t=1990}^{2022} P_{\text{country}}(t)}{\sum_{t=1990}^{2022} P_{\text{global}}(t)} \tag{1}$$

where $E_{\text{global}}$ is the global emissions, $E_{\text{country}}$ is the national emissions, $P_{\text{global}}$ is the global population, and $P_{\text{country}}$ is the national population—all in the year, $t$.

In line with Matthews[18], we argue that countries accumulate a carbon debt over time that is equal to the difference between the country's historic emissions and a per-capita allocation of global historic emissions. However, in contrast to Matthews[18], who uses an annual per capita allocation for historic emissions, we use equal cumulative per capita emissions also for the historic period. This choice is made to reduce the impact of individual years on the results given the close-to-linear relationship between cumulative emissions of CO$_2$ and global warming—the transient climate response to cumulative CO$_2$ emissions (TCRE)[2].

An alternative formulation would be to not distinguish between carbon debt and excessive carbon claims and estimate a country's share in the carbon budget from a historic starting year, such as 1990, combining historic emissions with the remaining global carbon budget. Robiou du Pont et al.[22] suggest such a formulation for the equal cumulative per capita emissions allocation principle. However, all countries have agreed to limit global warming and reduce emissions as of the adoption of the United Nations Framework Convention on Climate Change in 1992[8]. Hence, the comparatively higher emissions allocated to the historic period in our method is a joint failure of the parties to the UNFCCC. While one could argue that this is not entirely fair, the alternative—to calculate the equal share for a carbon budget that covers all years over the period 1990–2070—would result in future populations being given the right to (and responsibility for) historic emissions. Indeed, we included this method in our sensitivity analyses, see Supplementary Note 1 for details on the methodological choices tested and Supplementary Figs. 4, 5, and 6 for the results.

Similar to Fyson et al.[20] and Ganti et al.[19], we choose to estimate the carbon debt from 1990, when the first IPCC assessment report signified an emerging global scientific consensus and the negotiations on the UNFCCC started. However, this assumption is normative[10,19,20] and we have therefore also included other starting years—1960, 1980, and 2000—in the sensitivity analysis (Supplementary Fig. 4). We also provide estimates of countries' carbon debts using consumption-based emission accounting in comparison to territorial emission accounting (Supplementary Fig. 7) based on available data (i.e. for 1990–2021 and excluding Norway and Gambia).

### Excessive carbon claims
Excessive carbon claims estimate the difference between a country's expected future cumulative emissions, $CE_{\text{country}}$, based on pathways where it achieves its NDC and NZT, and an equal cumulative per capita allocation of the remaining global carbon budget (CB) for each respective country, also based on Robiou du Pont et al.'s[22] approach for allocating a carbon budget based on equal cumulative per capita emissions, see Eq. (2). The robustness of the choice of allocation principle is tested by estimating the additional carbon accountability also using alternative allocation principles based on previous studies[24,25]. The mathematic formulations for these alternatives are provided in Supplementary Note 1 and results are provided in Supplementary Figs. 4, 5, and 6.

$$\text{Excessive carbon claims} = CE_{\text{country}} - CB \cdot \frac{\sum_{t=2023}^{2070} P_{\text{country}}(t)}{\sum_{t=2023}^{2070} P_{\text{global}}(t)} \tag{2}$$

The cumulative emissions, $CE_{country}$, are estimated by creating simplified pathways towards an NZT for each country under the assumption that countries' NDCs and NZTs are achieved. It should be noted that this may be an optimistic assumption. Rogelj et al.[7] assessed whether the targets are legally binding if credible plans for implementation are in place, and near-term policies contribute to reducing emissions. If only realistic NZTs are accounted for, on top of current policies, "warming is projected to increase to 2.4 °C by 2100 (range due to emissions projection uncertainties: 1.7–3.0 °C)"[7].

The simplified pathways develop linearly from the country's 2022 emission level to its 2030 emission target, as reported in the country's NDC and quantified by Climate Action Tracker (CAT)[33], after which the pathway continues from 2030 until zero at the NZT year as communicated by the country (Fig. 1a). CAT gives a range of what the NDC means in greenhouse gas emission reductions, excluding land use, land-use change and forestry (LULUCF). Since we only analyse the fossil $CO_2$ budget, we apply the same percentage reduction on fossil $CO_2$ emissions as was observed for total greenhouse gas emissions. Note that other greenhouse gases are expected to form a large part of the residual greenhouse gas emissions when the net-zero target is reached. Hence, our assumption is not necessarily more ambitious than the stated NZT. Our study focuses on only $CO_2$ emissions given the TCRE–the close-to-linear relationship between cumulative emissions of $CO_2$ and global warming[2]. However, the size of any global carbon budget depends on assumptions of future emissions of other greenhouse gases, see Table 5.

Since our calculations exclude emissions and removals by LULUCF, we deduct 25 $GtCO_2$ (Table 5) for emissions occurring within LULUCF from 2023 up until 2058, when net emissions in LULUCF turn negative, based on the IPCC's Illustrative Mitigation Pathway for renewables (IMP-Ren)[34]. When a range of emission reductions is given for an NDC, the mean value in the range is used. For countries with a conditional and unconditional definition of their NDC (i.e. where the country makes their emission reductions conditional on support from other countries), the unconditional one (higher emissions) is used. For countries without stated net-zero targets (i.e. Egypt, Iran, Kenya, Mexico, Morocco, Philippines and the United Arab Emirates), we assume 2070 as the targeted year. We assume that all definitions of net-zero targets result in zero emissions of fossil $CO_2$ at the target year. For 'Rest of the World', including international aviation and shipping, we assume flat emissions until 2030, and a linear reduction from 2030 to net zero in 2070 (Supplementary Table 2).

The cumulative emissions for a country during the period 2023–2070 can be estimated using Eq. (3), where $ET_{country}$ in 2030 is the emission level in 2030 if the country's NDC is achieved and $t_{\text{net-zero}}$ denotes the communicated year for achieving net-zero emissions.

$$CE_{country} = \frac{E_{country}(2022) + ET_{country}(2030)}{2} \cdot (2030 - 2021) \\ - E_{country}(2022) + \frac{ET_{country}(2030)}{2} \cdot (t_{net-zero} - 2031) \tag{3}$$

Again, we assume here that countries achieve both their NDC targets for 2030 and their NZT pledges; the additional national carbon accountability would be larger otherwise (for countries with a positive additional accountability).

The total excessive carbon claims, $TECC_{country}$, is the sum of a country's carbon debt and excessive carbon claims, see Eq. (4).

$$TECC_{country} = \text{Carbon debt}_{country} + \text{Excessive carbon claims}_{country} \tag{4}$$

In this study, calculations are conducted for carbon budgets for both 1.5 °C with 50% probability and 2 °C with 83% probability[3], but our method can be used with any carbon budget.

## Additional carbon accountability

We define additional carbon accountability as each country's responsibility to mitigate $CO_2$ emissions or remove $CO_2$ in addition to its NDC and NZT pledges to stay within its share of a specific global carbon budget allocated based on equal cumulative per capita emissions and accounting for carbon debts based on the same method. The additional carbon accountability is conceptualised as follows. We allocate zero accountability to all countries whose total excessive carbon claim is negative, i.e. the sum of their carbon debt and future excessive carbon claims, as calculated in Eqs. (1–3). The emission allowances that these countries refrain from using together build up a global emission allowances pool, EAP, that reduces the global (net) excessive carbon claims. The global excessive carbon claims are therefore smaller compared to if all these low-emission countries were planning to emit according to an equal cumulative per capita allocation. The additional carbon accountability, ACA, is then estimated iteratively by redistributing those countries' unused emission allowances until the pool of emission allowances is exhausted and no country is assigned a negative additional carbon accountability. The iterative process is mathematically described in Eqs. (5–7):

$$EAP_n = -1 \cdot \sum_{i \in \{\text{all countries}|ACA_{i,n-1}<0\}} ACA_{i,n-1} \tag{5}$$

$$EAP_{country,n} = EAP_n \cdot \frac{\sum_{t=2023}^{2070} P_{country}(t)}{\sum_{i \in \{\text{all countries}|ACA_{i,n-1}>0\}} \sum_{t=2023}^{2070} P_i(t)}, \text{ and} \tag{6}$$

$$ACA_{country,n} = \begin{cases} ACA_{country,n-1} - EAP_{country,n} & ACA_{country,n-1}>0 \\ 0 & ACA_{country,n-1}<0 \end{cases} \tag{7}$$

where $n = 1, 2, 3, \ldots$ until the condition $EAP_n = 0$ is met. The additional carbon accountability, ACA, for a country is equal to its total excessive carbon claim in the first iteration, $ACA_{country,n=0} = TECC_{country}$. The additional carbon accountability is estimated by subtracting the equal cumulative per capita share of the emission allowances pool, $EAP_{country,n}$, from the country's additional carbon accountability in the previous iteration, $ACA_{country,n-1}$ (Eq. 7). The equal cumulative per capita share of the pool is estimated as in Eq. (2), but where the share of the cumulative future population is estimated based on the sum of countries that have positive additional carbon accountability in the previous iteration, see Eq. (6). Also similar to Eq. (2), the robustness of the choice of allocation principle to estimate a country's share in the emission allowances pool, $EAP_{country,n}$, is tested by also using alternative principles for redistributing the EAP. The mathematic formulations for these alternatives are provided in Supplementary Note 1 and results are provided in Supplementary Figs. 4, 5, and 6. To calculate the additional carbon accountability per capita, we used the country's average population for future years (2023–2070).

## Additional costs for mitigation and CDR

Finally, we estimate the cost for each country to deliver additional emission reductions or carbon dioxide removal (CDR) in line with their accountability. We used the ability to pay metrics GDP per capita, which is a common metric for capabilities[16]. We apply a least-cost principle and assume for simplicity that the average cost for cost-effective measures is 150 USD per ton of $CO_2$ mitigated or removed; this is within the estimated cost range of most CDR methods (see paragraph C.3.5 by the IPCC[35]), when countries have already employed all low-cost measures for achieving their NDCs and NZTs. For example, one review concludes that the cost range is 100–200 USD/ton for BECCS and 100–300 USD/ton for DACCS[36] while other assessments suggest lower cost ranges[37]. We assume the technologies for CDR to be internationally standardised and therefore choose to measure the ability to pay in current USD. However, since the global potential for

BECCS is limited by area and DACCS is limited by scaling and costs[38], a large part of countries' additional carbon accountability may be met by mitigation.

## Data sources

The impacts of countries' NDCs on their emission pathways are based on the estimates by Climate Action Tracker (CAT)[33]. We include all 37 countries (the European Union is treated as an aggregated unit) for which CAT has quantified targets for 2030 (Supplementary Table 2). Those are Argentina, Australia, Brazil, Canada, Chile, China, Colombia, Costa Rica, Egypt, Ethiopia, the European Union's 27 Member States combined, Gambia, India, Indonesia, Iran, Japan, Kazakhstan, Kenya, Mexico, Morocco, New Zealand, Nigeria, Norway, Peru, Philippines, Russia, Saudi Arabia, Singapore, South Africa, South Korea, Switzerland, Thailand, Türkiye, United Arab Emirates, United Kingdom, United States, and Vietnam. Bhutan is excluded due to the complexity of it already reaching net zero with LULUCF measures. For the 'Rest of the World', we assumed flat emissions until 2030 and then a linear decrease to net zero in 2070. We also assumed net-zero 2070 for the countries which have not made such a pledge.

Estimates on the country-level data for fossil $CO_2$ emissions in historic years (1990–2022) are based on the Global Carbon Project[39] and population data from the UN[40]. We use the estimated and projected population data for 1990-2070 from the 2022 Revision of the United Nations World Population Prospects[40] (median reproduction scenario), resulting in a world population of 10.3 billion in 2070.

In 2022, the selection of countries covered 6.0 billion people (75% of the global population), produced 93% of the global GDP, and caused fossil $CO_2$ emissions of 32.9 billion tonnes (89 % of global fossil $CO_2$ emissions). The remaining fossil $CO_2$ emissions are allocated to the 'Rest of the World' category, covering 4.2 $GtCO_2$ (11% of global fossil $CO_2$ emission) of which 1.1 $GtCO_2$ (3% of global fossil $CO_2$ emission) is due to international transportation.

Data for gross domestic product (GDP) per capita (current USD, 2021) was downloaded from the World Bank[41]. We used the World Bank classification of countries by income level[42]. Data for military expenditures (also in current USD, 2021) is from Stockholm International Peace Research Institute[25] for all countries except the European Union, which is sourced from Eurostat[43].

## Data availability

The data used, the calculations for the main case in Excel format, and detailed results for the sensitivity cases are provided in the data repository Zenodo[44], https://doi.org/10.5281/zenodo.10171891.

## Code availability

The code is written in R and available in the GitHub repository, https://github.com/morfeldt/AdditionalCarbonAccountability. The final code used for generating the results for this article has been preserved in the data repository Zenodo[45], https://doi.org/10.5281/zenodo.13815616.

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

## Acknowledgements

We thank reviewers for valuable comments. This research has been financed, for T.H., R.H., I.F., and M.K., by a grant to FAIRTRANS from Mistra [DIA 2019/28] and from Formas via the national research programme on climate (2021-00416). J.M.: work has been supported by Mistra Carbon Exit financed by Mistra, the Swedish Foundation for Strategic Environmental Research.

## Author contributions

T.H., R.H., J.M., M.K., and I.F. contributed jointly to conceptualizing the research and writing the article. R.H. and J.M. performed the computational analysis and J.M. visualized the results.

## Funding

## Competing interests
R.H. owns Marginal Carbon AB (consultancy in climate policy and carbon removal), which is a minority owner of CDR.fyi (carbon removal market data provider). R.H. holds paid advisory roles with Milkywire Climate Transformation Fund and Carbon Gap, and unpaid roles with the EU Expert Group on Carbon Dioxide Removal and SBTi Expert Advisory Group. These engagements have not influenced the research presented. Other authors declare no competing interests.
