## [Peer Review File · Nature Communications]

REVIEWER COMMENTS

Reviewer #1 (Remarks to the Author):

This is an interesting paper which can make a useful contribution to the literature on fair-share approaches to climate debt and climate mitigation obligations. However, several weaknesses need to be addressed before the manuscript is suitable for publication.

The main problem is that the author divides the assessment period between past (1990-2021) and future (2022-2070). For the former period ("national carbon debts"), national emissions are assessed against an equal per capita share of historical global emissions during the period. For the latter period ("excessive carbon claims"), nations' planned emissions are assessed against an equal per capita share of the remaining carbon budget for 1.5C. The method for the latter period is sound but I see no good justification for the approach taken for the former period, as the fair-shares are overdetermined by the scale of global emissions during the period, which are in turn overdetermined by the emissions of high-emitting countries. This approach seems inconsistent with the principle of the carbon budget and falls foul of the equity-based fair-shares principles established in previous literature (see below).

Instead, it seems much more reasonable to use something like the latter method consistently across the whole assessment period. This can be done by taking the remaining carbon budget for 1.5C as of 2022, and adding observed emissions for 1990-2021, to derive a carbon budget for 1.5C for the period from 1990. Nations' observed emissions (and observed emissions plus planned emissions) should then be measured against equal per capita shares of the carbon budget from 1990 (in other words, using national population as a share of global population over the analysis period, as the authors have done).

This is the approach that was taken by Hickel (2020) and Fanning and Hickel (2023) (<https://www.sciencedirect.com/science/article/pii/S2542519620301960>; <https://www.nature.com/articles/s41893-023-01130-8>), the latter of which (in Nature Sustainability) represents what is to my knowledge the most recent contribution to the literature on carbon budget fair-shares and should be engaged. Fanning provides country-level results online here: <https://goodlife.leeds.ac.uk/related-research/atmospheric-appropriation/>, with a baseline option for 1992. With the 1992 baseline, it seems that results differ substantially from what is obtained in the present paper, with respect to the historical period. For instance, Fanning's website indicates that China is still within its fair-share of the 1.5C carbon budget (with emissions through 2019).

A second major question has to do with the selection of the baseline year, 1990. This approach is justified, however I think it would be worthwhile to include an earlier baseline year for comparison - such as 1960, which is used in the papers cited above and others. The results of such an exercise should be mentioned in the main text and can be represented more fully in the SI.

-In table 1, the colour coding does not seem necessary, as the countries are already ranked by GDP per capita. Perhaps better to use a "traffic light" colour coding, for instance to indicate those with carbon debt and excessive carbon claims in red, those with only one or the other in yellow, and those with neither (both negative) in green.

-line 220, the \$100 figure seems low, and the justification for selecting this figure seems weak. It seems to me that the price should be selected based on abatement costs in IPCC AR6 scenarios consistent with 1.5C. I believe these are considerably higher than \$100. In Fanning and Hickel (SI table 3) they indicate \$100-500 depending on the year.

-the linear future pathways seem a bit odd when most studies use Raupach curves or similar. For robustness, at least one alternative pathway shape should be tested so that we know how sensitive the results are to the chosen pathways shape.

-I feel that some important references are missing. For example:

Pickering, J. & Barry, C. On the concept of climate debt: its moral and political value. *Crit. Rev. Int. Soc. Polit. Phil.* 15, 667–685 (2012).

Narain, S. & Riddle, M. in *Reclaiming Nature: Environmental Justice and Ecological Restoration* (eds Stanton, E. et al.) 401–414 (Anthem Press, 2007)

Fair Shares: A Civil Society Equity Review of INDCs (CSO Review, 2015); <https://www.equityreview.org/>

Fanning, A. L., O'Neill, D. W., Hickel, J. & Roux, N. The social shortfall and ecological overshoot of nations. *Nat. Sustain.* 5, 26–36 (2022).

Holz, C., Kartha, S. & Athanasiou, T. Fairly sharing 1.5: national fair shares of a 1.5 °C-compliant global mitigation effort. *Int. Environ. Agreem.* 18, 117–134 (2018).

Sultana, F. Critical climate justice. *Geogr. J.* 188, 118–124 (2022).

Jayaraman, T. & Kanitkar, T. Deconstructing declarations of carbon neutrality. *Third World Resurgence* 347, 11–13 (2021).

Klinsky, S. et al. Why equity is fundamental in climate change policy research. *Glob. Environ. Change* 44, 170–173 (2017).

Vanderheiden, S. *Atmospheric Justice: A Political Theory of Climate Change* (Oxford Univ. Press, 2008); <https://doi.org/10.1093/acprof:oso/9780195334609.001.0001>

Reviewer #2 (Remarks to the Author):

The paper studies the gap between countries' targets (nationally determined contributions (NDCs) and net-zero targets) and a fair carbon budget computed from an equal cumulative emissions per capita approach. To measure this gap, the authors define the 'Fair National Carbon Accountability Indicator'.

The indicator measures for each country how much the carbon budget from countries' targets deviates from the fair carbon budget. Some countries do not plan to use up their fair carbon budget, the excess is then also distributed to countries whose targets lack behind the fair budget and it is distributed in proportion to how much they deviate. In addition, the paper studies the ability to pay of countries for this accountability.

The paper is very well written, and the analysis of the defined indicator is very clear.

However, while it can be important to define and study such indicators, it is not clear why the indicator is defined in this way:

1) Why are only CO₂ emissions considered?

2) Why is the 'Fair National Carbon Accountability Indicator' discounted in proportion to the fair budget that is not used up by some countries. This means that countries with a larger gap between targets and the fair budget, get a higher discount, which seems flawed. It seems the 'Total excessive claims' (defined in the paper) would be a better indicator since it does not have this flaw.

3) What is the novelty of the indicator especially compared to the 'Total excessive claims'? The 'Fair National Carbon Accountability Indicator' of each country is related to the 'Total excessive claims' by the same factor for all countries, so what new information does it provide?

Overall, the indicator should be explained and justified more.

Reviewer #3 (Remarks to the Author):

Key results

The study proposes a new approach to fair distribution of a remaining carbon budget and burden sharing of climate change mitigation, with the aim of linking top-down budget approaches to in their view bottom-up nationally determined contributions. The study combines an estimate of responsibility for past emissions and projected national emissions as compared to the budget under a cumulative equal per capita approach to determine the degree to which a country has used or will in the future use more than their fair share of remaining emissions to remain under a stringent climate target (1.5 degrees, with an additional analysis for 2 degrees found in extended data). The upshot is an indicator which tries to merge up to now somewhat separate but related concepts (fairness budgets vs NDCs) to provide an alternative approach to informing as to adequacy of national climate plans and need for additional mitigation and carbon removal. The indicator's use in identifying groups of countries which can/cannot fully account for previous and future planned emissions via mitigation alone, versus requiring CDR is a novel application. Overall I think the concept is an interesting one, and to my knowledge represents a unique and novel approach to try to bridge NDCs and other budget allocation methods.

I'd like to thank the authors for their work; it was an enjoyable read and I do think the idea is a novel one and could be a nice addition to the field. I have some comments which I hope can be helpful to the authors in further steps of refining the work for publication, which follow below.

Validity

In terms of robustness of the data interpretation and conclusions, I think the point essentially hinges on the assumption that "differentiated national responsibility... is a more reasonable and fair principle for

allocating accountability.” (Discussion, beginning line 250). The validity of the results relies on the NDCs being a “bottom-up approach to fairness” (line 67) but no argument is made to this effect. The study cited (Winkler et al. 2018) does not seem to support this, stating (in the abstract) “Broadly, we find that most claims to equity are either unsubstantiated or drawn from analysis by in-country experts.” The work asserts that the indicator is an improvement on previous fairness approaches, but one could argue (as one of the citations in the study (Williges et al, 2022) does) that many NDCs implicitly rely on grandfathering, which is inconsistent with fair allocation of remaining budgets. To the author’s credit, they do acknowledge this potential issue, but do not resolve it. The benefit of incorporating NDCs as in this work would seem more related to political acceptability or feasibility arguments, but the focus of the work is on the asserted Fair indicator. There are two other instances (more minor in nature) where the normative implications are referenced but not made clear. Line 263, and Methods line 349. Again these are less vital to the overall argument, but follow a theme of the work. The indicator is “Fair National Carbon Accountability”, but there is little argumentation in the study that it is in fact fair.

Significance

As mentioned previously, I do think the new indicator is interesting and could be a potential way to bridge NDCs and top-down fairness budget allocations. I think in its current form, the study would be of interest to the field, but I struggle to assess to what degree it would be significant, and the relevance to related fields. I think the authors could improve upon the significance by placing their new indicator in context – as it is, the Carbon Debts and Excessive Carbon Claims section presents a lot of results, but doesn’t really contextualize how their indicator performs relative to other means of allocating budgets. The study would do well to help the reader understand that beyond the method being new and better, it leads to different conclusions. As it is the reader would have to determine this on their own, comparing to recent works on similar topics etc. It would be helpful to make the case of significance and novelty if the authors illustrate their method is an improvement in terms of fairness as discussed above (or minimize the fairness argument and focus on more practical considerations) and drive home the novelty of results.

Data and methodology

Validity of the approach: The calculations of national carbon debt, excessive carbon claims etc. are all based in previous allocation approaches. As mentioned previously, there are some value judgements made in e.g. Step 1 calculating national carbon debt (line 349, and also the choice of using Robiou du Pont’s approach and not Matthews (line 343-344) but these can be justified. (It would benefit the reader if the authors would clarify why steps were taken, e.g. when they state it’s a ‘value judgement’ etc. Not a long argument, but perhaps a sentence or two of elaboration.

In terms of data quality: Limiting the study to the countries of the CAT database seems fine, but I would caution the authors to recheck their data, perhaps it is out of date. I did not have time to check all countries, but when looking at Argentina, in the extended data .xls file (sheet “2030 emission estimates”) the climate action tracker estimate is given as 315.9 MtCO_{2e} in 2030; the 2030 estimate as of January 2024 appears to be 398 Mt, unless I’ve found a different set of scenario estimates.

Quality of presentation: Here I think some improvements could be made, Particularly w.r.t. Figure 1 and all Tables.

All figures: Improve readability – text is quite small.

Figure 1: Panel labels missing; 1b is difficult to gain any meaning from, since I have to work hard to figure out which countries correspond to which lines. Better would be to do similar to panel A and include the country names on the left, in between the axis and the start of the line. Panel B is meant to illustrate highest carbon claims per capita for high-income countries – perhaps keep the income level coloration from panel A. Caption of Figure 1: why do you only show countries with claims higher than 15,000 Gt (Gt? This must be a different unit)? It seems an arbitrary number, try and justify differently perhaps.

Table 1: This table seems far too big and busy – I can understand wanting to convey all results for readers, but they run the risk of getting lost in the table and details. I would recommend trying to trim down the number of countries you list here, tied to the countries you discuss at more length in the text. Colors to divide into income groups – what is the rationale for the divisions? Instead of colors, probably best to try and separate them with a row of merged cells, indicating e.g. “High income”, “upper middle” etc. to avoid readability issues if/when printing. Explain the rationale for the division more specifically in the table caption text.

Figure 2: I quite like this figure and think it is helpful to convey your results and get an idea of where all the countries stand etc. It may be helpful to emphasize the zero lines, and perhaps label the quadrants / areas indicating as you say in the paper groups of high-income countries where their accountability is higher than their planned emissions, and conversely the larger emissions than accountabilities (so different ways to fulfill their obligations) – the reader could already spot that from your chart. The colors could be more helpful (yellow is hard to read on my screen at least). Would it help your scaling issue (e.g. China) if you put everything in per capita terms? This may then make the comparison of claims and debt impossible though, given the different numbers, so maybe not. While the exploded view (left panel) is useful, it seems a shame to waste so much space on the right panel, as its quite hard to read the country names. But, I can understand the necessity, and the way the exploded view is done is quite helpful for the reader.

Table 2: Needs a more descriptive table caption (at least a full sentence) but I wondered as to the value of this table when reading the paper. It is only referred to once or twice, in two sentences discussing G7 and BRICS, but the disaggregated results do not seem so interesting – I as reader can check in the main table if I want the details. Or if there’s further importance of this breakdown, it needs more elaboration in the table caption and text.

Analytical approach

Strength of analytical approach – I believe my comments above have generally covered this area as well.

Suggested improvements

Again, see my above comments, but two main improvements:

1. Strengthen the discussion of fairness, and make an argument that the proposed indicator is indeed fair (as per discussion in Validity above)
2. Provide a means of comparison for the new indicator – place it in related / relevant literature /

estimates. It is hard to judge the impact of the indicator and this study without some reference for reader to compare with.

Clarity and context

See above – the study, while referencing previous work that provides overviews as to fairness considerations, and the need to address normative aspects of carbon budgets explicitly, does not do so to a large degree, and the work could be improved by adding this additional context (as well as the context in terms of how the results compare).

References

No major issues.

REVIEWER COMMENTS

Reviewer #1 (Remarks to the Author):

This is an interesting paper which can make a useful contribution to the literature on fair-share approaches to climate debt and climate mitigation obligations. However, several weaknesses need to be addressed before the manuscript is suitable for publication.

The main problem is that the author divides the assessment period between past (1990-2021) and future (2022-2070). For the former period ("national carbon debts"), national emissions are assessed against an equal per capita share of historical global emissions during the period. For the latter period ("excessive carbon claims"), nations' planned emissions are assessed against an equal per capita share of the remaining carbon budget for 1.5C. The method for the latter period is sound but I see no good justification for the approach taken for the former period, as the fair-shares are overdetermined by the scale of global emissions during the period, which are in turn overdetermined by the emissions of high-emitting countries. This approach seems inconsistent with the principle of the carbon budget and falls foul of the equity-based fair-shares principles established in previous literature (see below).

Instead, it seems much more reasonable to use something like the latter method consistently across the whole assessment period. This can be done by taking the remaining carbon budget for 1.5C as of 2022, and adding observed emissions for 1990-2021, to derive a carbon budget for 1.5C for the period from 1990. Nations' observed emissions (and observed emissions plus planned emissions) should then be measured against equal per capita shares of the carbon budget from 1990 (in other words, using national population as a share of global population over the analysis period, as the authors have done).

This is the approach that was taken by Hickel (2020) and Fanning and Hickel (2023) (<https://www.sciencedirect.com/science/article/pii/S2542519620301960>; <https://www.nature.com/articles/s41893-023-01130-8>), the latter of which (in Nature Sustainability) represents what is to my knowledge the most recent contribution to the literature on carbon budget fair-shares and should be engaged. Fanning provides country-level results online here: <https://goodlife.leeds.ac.uk/related-research/atmospheric-appropriation/>, with a baseline option for 1992. With the 1992 baseline, it seems that results differ substantially from what is obtained in the present paper, with respect to the historical period. For instance, Fanning's website indicates that China is still within its fair-share of the 1.5C carbon budget (with emissions through 2019).

Our response

Thanks for many valuable comments and for highlighting that our approach differs from previous studies in the field. We have made substantial revisions and enclose a separate document with track changes. We see our estimates of both past and future emissions as important and in doing so, we consider the approach to be consistent "with the principle of the carbon budget" and being grounded in "equity-based fair-shares principles". On Line 430-442 we added:

In line with Matthews (2016), we argue that countries accumulate a carbon debt over time that is equal to the difference between the country's historic emissions and a per-capita allocation of global historic emissions. Since the adoption of the UNFCCC in 1992, all countries have agreed to limit global warming and reduce emissions. Hence, we further argue that the comparatively higher emissions allocation to the historic period in our method is reflecting the joint failure of the parties to the UNFCCC, since this is what has happened. If we calculate the equal share 1990-2070, then future populations are given the right to (and responsibility for) past emissions.

An added benefit of our approach is enabling a disaggregated analysis that can highlight the interplay between a country's carbon debt and excessive carbon claims (e.g., Table 3)

With that said, we agree with the reviewer that the methods used in previous studies should be tested and compared with our new approach. We have included the equal cumulative emissions per capita during 1990-2070 in our sensitivity analysis, see Supplementary Figures 7 and 8. We note that shifting to this approach has a fairly low impact on the results (below $\pm 10\%$), except for Chile and Thailand whose accountability increases notably but from low levels per capita. Further, shifting to this approach would result in reduced additional carbon accountability for many high-income countries (see Line 265-281).

In regards to the comparison with Fanning and Hickel (2023), they use consumption-based emissions, and assume reductions of "emissions at a constant rate using a simple exponential function" from 2020 to 2050, implying the same net-zero year for all countries. Hence, it is not relevant for their study to account for territorial targets such as the NDCs and NZTs. This means that our results, which use territorial emissions accounting and estimate the additional accountability on top of achieving a country's NDC and NZT, are not comparable with their study.

A second major question has to do with the selection of the baseline year, 1990. This approach is justified, however I think it would be worthwhile to include an earlier baseline year for comparison - such as 1960, which is used in the papers cited above and others. The results of such an exercise should be mentioned in the main text and can be represented more fully in the SI.

Our response

Thanks for raising this. We agree that an analysis of how sensitive the results are to the choice of baseline year is missing. Hence, we have added a sensitivity analysis in Supplementary Figure 6, where we show the impact of assuming a baseline year of 1960, 1980, or 2000 in comparison to the main case, where 1990 is assumed.

Generally, we note that an earlier baseline year increases the Additional Carbon Accountability for high-income countries and the opposite for a later baseline. To highlight two examples, the United States has an Additional Carbon Accountability of 180, 155, and 120 GtCO₂ for baseline years of 1960, 1980 and 2000, respectively, which can be compared to 140 GtCO₂ in the main case. China has an Additional Carbon

Accountability of 103, 145, and 210 GtCO₂ for baseline years of 1960, 1980 and 2000, respectively, which can be compared to 175 GtCO₂ in the main case. (Line 252-263)

-In table 1, the colour coding does not seem necessary, as the countries are already ranked by GDP per capita. Perhaps better to use a "traffic light" colour coding, for instance to indicate those with carbon debt and excessive carbon claims in red, those with only one or the other in yellow, and those with neither (both negative) in green.

Our response

We have removed the colour codes in Table 1 as suggested and instead indicate the economic development categories in the table, as suggested by the Reviewer 3.

-line 220, the \$100 figure seems low, and the justification for selecting this figure seems weak. It seems to me that the price should be selected based on abatement costs in IPCC AR6 scenarios consistent with 1.5C. I believe these are considerably higher than \$100. In Fanning and Hickett (SI table 3) they indicate \$100-500 depending on the year.

Our response

Thanks for raising this issue. We agree that the figure seems low and have changed it to 150 USD/ton. This is within the estimated cost range of most CDR methods (IPCC AR6 WG3). The following text has also been added in the Methods section:
"For example, one review concludes that the cost range is 100-200 USD/ton for BECCS and 100-300 USD/ton for DACCS (Fuss et al. 2018)." (Line 547-548)

Fuss, S. et al. (2018). Negative emissions—Part 2: Costs, potentials and side effects. *Environmental research letters*, 13(6), 063002.

-the linear future pathways seem a bit odd when most studies use Raupach curves or similar. For robustness, at least one alternative pathway shape should be tested so that we know how sensitive the results are to the chosen pathways shape.

Our response

While we understand the reviewer's concern about a linear pathway not being a plausible development, we argue that a linear pathway is more transparent and representative of the targets set by policy-makers than a Raupach curve. A Raupach curve is based on the emission levels of the base year and two parameters: a rate of increase in emissions in the short-term (r), and a rate of decrease in emissions in the long-term (m) (Raupach et al. 2014).

The two parameters can be estimated based on an assumption of the increase of emissions in the short-term and a carbon budget, but this is not how countries' policy targets generally are formulated. Hence, assumptions in addition to the targets set by policy-makers need to be made by us researchers, and it would be difficult to fit the curve to the 2030 emission target and the year for reaching zero. Using a Raupach curve would therefore reduce the useability of our method for practitioners

(governments, NGOs, etc.). In fact, a Raupach curve never reaches zero and is therefore not useful for representing the policy targets that we analyze.

Furthermore, we are primarily interested in the cumulative emissions and due to the assumptions needed to form a Raupach curve, it could easily be manipulated to result in equal cumulative emissions as the linear curve (e.g., the linear curve for planned emissions for the US results in cumulative emissions of 62.7 GtCO₂ over 2023-2070 and by assuming $r = 0.04$ and $m = 0.17295$ we can achieve that same level of cumulative emissions. Hence, there is no added benefit of using Raupach curves.

-I feel that some important references are missing. For example:

Pickering, J. & Barry, C. On the concept of climate debt: its moral and political value. *Crit. Rev. Int. Soc. Polit. Phil.* 15, 667–685 (2012).

Narain, S. & Riddle, M. in *Reclaiming Nature: Environmental Justice and Ecological Restoration* (eds Stanton, E. et al.) 401–414 (Anthem Press, 2007)

Fair Shares: A Civil Society Equity Review of INDCs (CSO Review, 2015);
<https://www.equityreview.org/>

Fanning, A. L., O’Neill, D. W., Hickel, J. & Roux, N. The social shortfall and ecological overshoot of nations. *Nat. Sustain.* 5, 26–36 (2022).

Holz, C., Kartha, S. & Athanasiou, T. Fairly sharing 1.5: national fair shares of a 1.5 °C-compliant global mitigation effort. *Int. Environ. Agreem.* 18, 117–134 (2018).

Sultana, F. Critical climate justice. *Geogr. J.* 188, 118–124 (2022).

Jayaraman, T. & Kanitkar, T. Deconstructing declarations of carbon neutrality. *Third World Resurgence* 347, 11–13 (2021).

Klinsky, S. et al. Why equity is fundamental in climate change policy research. *Glob. Environ. Change* 44, 170–173 (2017).

Vanderheiden, S. *Atmospheric Justice: A Political Theory of Climate Change* (Oxford Univ. Press, 2008); <https://doi.org/10.1093/acprof:oso/9780195334609.001.0001>

Our response

Thank you for these references! We have included two of them in the manuscript (Holz et al. 2018 and Klinsky et al. 2017):

“Such a two-step approach to fairness is not unique. For example, Holz et al.²⁴ have suggested “dual obligations” for countries by first meeting their own national targets and then allocating additional obligations to countries with excessive emissions (large cumulative emissions) and high capacity (economic wherewithal).” (Line 349-352)

“ Governments’ perceptions of what is “fair enough” are manifested in negotiations and commitments like NDCs (Klinsky et al. 2017)¹⁶” (Line 77-78)

Reviewer #2 (Remarks to the Author):

The paper studies the gap between countries’ targets (nationally determined contributions (NDCs) and net-zero targets) and a fair carbon budget computed from an equal cumulative emissions per capita approach. To measure this gap, the authors define the ‘Fair National Carbon Accountability Indicator’. The indicator measures for each country how much the carbon budget from countries’ targets deviates from the fair carbon budget. Some countries do not plan to use up their fair carbon budget, the excess is then also distributed to countries whose targets lack behind the fair budget and it is distributed in proportion to how much they deviate. In addition, the paper studies the ability to pay of countries for this accountability.

The paper is very well written, and the analysis of the defined indicator is very clear. However, while it can be important to define and study such indicators, it is not clear why the indicator is defined in this way:

1) Why are only CO₂ emissions considered?

Our response

Thank you for your many valuable comments. We have made substantial revisions and enclose a separate document with track changes. The reason for focusing on CO₂ emissions is the close-to-linear relationship between cumulative emissions of CO₂ and long-term global temperature change - a relationship called the transient response to cumulative carbon emissions (TCRE).

This relationship is what enables cumulative emissions to be a proxy for a country’s contribution to global warming and without it the concept of carbon budgets is not relevant. Other greenhouse gases of course also impact the global temperature and are important to reduce over time, but is harder to be described in terms of an explicit budget. Instead, the global carbon budget is adjusted to account for projected future emissions of other greenhouse gases.

In our case, the budgets applied for CO₂ imply “reductions in other greenhouse gases between 2020–2050, such as CH₄ (50 %) and N₂O (25 %).” (see caption of Table 5). To clarify this, we have now added under Method:

“Our study focuses on only CO₂ emissions given the TCRE - the close-to-linear relationship between cumulative emissions of CO₂ and global warming. However, the size of any CO₂ global carbon budget depends on assumptions of future emissions of other greenhouse gases, see Table 5.” (Line 479-482)

2) Why is the ‘Fair National Carbon Accountability Indicator’ discounted in proportion to the fair budget that is not used up by some countries. This means that countries with a larger gap between

targets and the fair budget, get a higher discount, which seems flawed. It seems the 'Total excessive claims' (defined in the paper) would be a better indicator since it does not have this flaw.

Our response

If we had used "total excessive carbon claims" as indicator we would assume that equal cumulative per capita emissions 1990-2070 is the "correct" interpretation of fairness. However, equating fairness with equal is only one possible normative choice. Our ambition is not to assert what is fair per se but to allocate additional accountability on top of what countries have already agreed to. Countries should of course restrain themselves from emitting more than they express in their own plans. This does not exclude reasons for high capability countries to assist developing countries in their goal achievement, on the contrary, which we discuss in relation to challenges illustrated in Table 4:

"We acknowledge the need to "provide financial resources to assist developing country Parties" (Article 9) and such assistance may be needed also for some middle-income countries, which have considerable Additional Carbon Accountability for the 1.5°C budget on top of the challenges to achieve their national targets (Table 4)." (Line 374-377)

In their NDCs and NZTs, high income countries have generally committed to faster phase-out of CO₂ (by 2050) than low or middle-income countries, which is the norm in the Paris Agreement. Our ambition is to build on and stimulate dialogue on how to improve the implementation of the Paris Agreement, not to replace it with a hypothetical norm. We have deleted "discount" in the text since we realise this was misleading:

"The sum of only positive total excessive carbon claims is 789 GtCO₂ but the global net sum of excessive carbon claims is 576 GtCO₂, since 14 countries plan to emit less than what follows from an equal cumulative per capita emission approach for 1990-2070. This corresponds to 213 GtCO₂ or 27% of the large emitters' total excessive carbon claims." (Line 184-188)

3) What is the novelty of the indicator especially compared to the 'Total excessive claims'? The 'Fair National Carbon Accountability Indicator' of each country is related to the 'Total excessive claims' by the same factor for all countries, so what new information does it provide?
Overall, the indicator should be explained and justified more.

Our response

Thank you for this comment which catalysed lots of discussions. First, we have changed the name of the indicator from "Fair" to "Additional" to clarify that this is additional to the accountability of achieving nationally determined targets. This also clarifies that the aim with the article is not to propose a "fair" allocation per se but to improve the fairness of the existing institution, the Paris Agreement. Second, the novelty is our ambition to combine a bottom-up analysis of countries' climate targets with a top-down allocation of accountability to close the mitigation gap.

We have clarified and justified the indicator and the aim of the article;

“This Additional Carbon Accountability indicator operationalises the CBDR-RC principle and complements previous research by combining bottom-up estimations of country emission pathways, based on pledges made by governments, with a top-down equal cumulative per capita approach. The ambition is to build on the Paris Agreement, being the internationally agreed climate treaty, and improve the fairness of its implementation by acknowledging common but differentiated responsibilities for past and future emissions.” (Line 103-108).

Finally, we justify the indicator with a new sensitivity analysis, where different allocation principles for the remaining carbon budget are compared. (Line 243-281)

Reviewer #3 (Remarks to the Author):

Key results

The study proposes a new approach to fair distribution of a remaining carbon budget and burden sharing of climate change mitigation, with the aim of linking top-down budget approaches to in their view bottom-up nationally determined contributions. The study combines an estimate of responsibility for past emissions and projected national emissions as compared to the budget under a cumulative equal per capita approach to determine the degree to which a country has used or will in the future use more than their fair share of remaining emissions to remain under a stringent climate target (1.5 degrees, with an additional analysis for 2 degrees found in extended data). The upshot is an indicator which tries to merge up to now somewhat separate but related concepts (fairness budgets vs NDCs) to provide an alternative approach to informing as to adequacy of national climate plans and need for additional mitigation and carbon removal. The indicator’s use in identifying groups of countries which can/cannot fully account for previous and future planned emissions via mitigation alone, versus requiring CDR is a novel application. Overall I think the concept is an interesting one, and to my knowledge represents a unique and novel approach to try to bridge NDCs and other budget allocation methods.

I’d like to thank the authors for their work; it was an enjoyable read and I do think the idea is a novel one and could be a nice addition to the field. I have some comments which I hope can be helpful to the authors in further steps of refining the work for publication, which follow below.

Validity

In terms of robustness of the data interpretation and conclusions, I think the point essentially hinges on the assumption that “differentiated national responsibility... is a more reasonable and fair principle for allocating accountability.” (Discussion, beginning line 250). The validity of the results relies on the NDCs being a “bottom-up approach to fairness” (line 67) but no argument is made to this effect. The study cited (Winkler et al. 2018) does not seem to support this, stating (in the abstract) “Broadly, we find that most claims to equity are either unsubstantiated or drawn from analysis by in-country experts.” The work asserts that the indicator is an improvement on previous fairness approaches, but one could argue (as one of the citations in the study (Williges et al, 2022) does) that many NDCs implicitly rely on grandfathering, which is inconsistent with fair allocation of remaining budgets. To the author’s credit, they do acknowledge this potential issue, but do not resolve it. The benefit of incorporating NDCs as in this work would seem more related to political acceptability or feasibility arguments, but the focus of the work is on the asserted Fair indicator.

Our response

Thank you for this important comment. We have made substantial revisions and enclose a separate document with track changes. First, we have changed the name of the indicator from “Fair” to “Additional” to clarify that this is additional to the accountability of achieving nationally determined targets. Hence, the title rhymes better with our ambition to combine feasibility and fairness. See also our answer to the comment above by Reviewer 2.

We do not argue that the future claims (NDCs and NZTs) are fair, on the contrary, the larger future carbon claims a high-emitting country has, the larger becomes its additional carbon accountability. However, we argue that the Paris Agreement is the internationally agreed climate institution and therefore it is a reasonable endeavour to improve the fairness of its implementation. We agree with Winkler, bottom-up approaches to fairness concern what the governments perceive as fair and that is only a start, although important in international negotiations (we added a ref on that, Line 77-78).

Indeed, our indicator allocates much more accountability to high-income countries than any other allocation principle (see new sensitivity analyses in Supplementary Figures 7+8), because we account for historical emissions. (Line 265-281)

We do acknowledge the risk that NDCs are similar to grandfathering but when the years for net-zero are considered, another pattern emerges:

“On the contrary, many large developed countries take the lead in mitigation, hence respecting Article 4.4 of the Paris Agreement. Indeed, we found that three middle-income countries in the BRICS+ group – Iran, Russia and China – have larger carbon claims (planned emissions) per capita than any of the G7 countries (Table 3; Supplementary Table 1).” (Line 232-236)

There are two other instances (more minor in nature) where the normative implications are referenced but not made clear. Line 263, and Methods line 349. Again these are less vital to the overall argument, but follow a theme of the work. The indicator is “Fair National Carbon Accountability”, but there is little argumentation in the study that it is in fact fair.

Our response

We no longer make claims of the indicator being fair per se. It’s unavoidable though to make normative choices and we have changed the passage around Line 263 to:

“The ambition of this article is to help enhance climate fairness in the real world. The calculations assume that countries are accountable to the mitigation targets they have voluntarily committed to within the frames of the Paris Agreement, regardless if these commitments are more ambitious than what a particular theoretical allocation principle would suggest is fair. This is of course, like all assumptions of fair allocations of carbon budgets, a normative assumption³¹. Another normative point of departure is the suggestion that the countries with an Additional Carbon Accountability are

collectively responsible to meet the global target. Such a two-step approach to fairness is not unique. For example, Holz et al.²⁴ have suggested “dual obligations” for countries by first meeting their own national targets and then allocating additional obligations to countries with excessive emissions (large cumulative emissions) and high capacity (economic wherewithal).” (Line 342-352)

Line 349 has been changed to:

Similar to Fyson et al.²⁰ and Ganti et al.¹⁸, we choose to estimate the carbon debt from 1990, when the first IPCC assessment report signified an emerging global scientific consensus and the negotiations on the UNFCCC started. However, this assumption is normative^{10,19,20} and we have therefore also included other starting years – 1960, 1980, and 2000 – in the sensitivity analysis.” (Line 444-448)

Significance

As mentioned previously, I do think the new indicator is interesting and could be a potential way to bridge NDCs and top-down fairness budget allocations. I think in it’s current form, the study would be of interest to the field, but I struggle to assess to what degree it would be significant, and the relevance to related fields. I think the authors could improve upon the significance by placing their new indicator in context – as it is, the Carbon Debts and Excessive Carbon Claims section presents a lot of results, but doesn’t really contextualize how their indicator performs relative to other means of allocating budgets. The study would do well to help the reader understand that beyond the method being new and better, it leads to different conclusions. As it is the reader would have to determine this on their own, comparing to recent works on similar topics etc. It would be helpful to make the case of significance and novelty if the authors illustrate their method is an improvement in terms of fairness as discussed above (or minimize the fairness argument and focus on more practical considerations) and drive home the novelty of results.

Our response

Thanks for this comment which inspired us to include a more detailed section on the impact of different assumptions on allocation principles.

First, we clarified the two-step nature of our analyses (accountability to achieve national targets and then additional accountability to stay within the global carbon budget).

Second, we have added a section on the robustness of the indicator, where we test different assumptions that have been made. We keep the framework that countries are first accountable for achieving their targets and that the indicator shows additional accountability for each country also for this sensitivity analysis, but change the allocation principle, the temperature target being achieved, and the assumed starting year for the historic responsibility. See Lines 245-291)

The results are presented in Supplementary Figures 6-8 and methods further described in Supplementary Notes.

Data and methodology

Validity of the approach: The calculations of national carbon debt, excessive carbon claims etc. are all based in previous allocation approaches. As mentioned previously, there are some value judgements made in e.g. Step 1 calculating national carbon debt (line 349, and also the choice of using Robiou du Pont's approach and not Matthews (line 343-344) but these can be justified. (It would benefit the reader if the authors would clarify why steps were taken, e.g. when they state it's a 'value judgement' etc. Not a long argument, but perhaps a sentence or two of elaboration.

Our response

We have elaborated the text where we acknowledge that some choices are necessarily normative, see answers above.

The following text was added on the choice of using Robiou du Pont et al.'s formulation:

"However, in contrast to Matthews'¹⁷ and van den Berg et al. who use an annual per capita allocation for historic emissions, we use equal cumulative per capita emissions also for the historic period. This choice is made to reduce the impact of individual years on the results given the close-to-linear relationship between cumulative emissions of CO₂ and global warming – the transient response to cumulative CO₂ emissions (TCRE"

And:

"...the comparatively higher emissions allocated to the historic period in our method is a joint failure of the parties to the UNFCCC. While one could argue that this is not objectively fair, the alternative – to calculate the equal share for a carbon budget that covers all years over the period 1990-2070 – would result in future populations being given the right to (and responsibility for) historic emissions. Indeed, we included this method in our sensitivity analyses, see Supplementary Note 1 for details on the methodological choices tested and Supplementary Figures 6, 7 and 8 for the results."

In terms of data quality: Limiting the study to the countries of the CAT database seems fine, but I would caution the authors to recheck their data, perhaps it is out of date. I did not have time to check all countries, but when looking at Argentina, in the extended data .xls file (sheet "2030 emission estimates") the climate action tracker estimate is given as 315.9 MtCO₂e in 2030; the 2030 estimate as of January 2024 appears to be 398 Mt, unless I've found a different set of scenario estimates.

Our response

We have updated the data with the latest assessments from CAT (Feb 2024).

Quality of presentation: Here I think some improvements could be made, Particularly w.r.t. Figure 1 and all Tables.

All figures: Improve readability – text is quite small.

Our response

Thank you for pointing out the small text. All figures have been improved with larger font sizes to improve readability.

Figure 1: Panel labels missing; 1b is difficult to gain any meaning from, since I have to work hard to figure out which countries correspond to which lines. Better would be to do similar to panel A and include the country names on the left, in between the axis and the start of the line. Panel B is meant to illustrate highest carbon claims per capita for high-income countries – perhaps keep the income level coloration from panel A. Caption of Figure 1: why do you only show countries with claims higher than 15,000 Gt (Gt? This must be a different unit)? It seems an arbitrary number, try and justify differently perhaps.

Our response

Thank you for these remarks. Both panels of the figure have been redone. Labels have been added to the left of each line to make it easier to see which country corresponds to which line. In the background, the historic development has been added to show how emissions have developed over time, but please note that it's future emissions that we focus on here, which is why the historic emissions are more transparent and behind the labels. We have also made a new selection of countries representing the four income levels considered in the tables. The same color coding is now used in all figures. We find it more interesting to highlight the differences between selected countries in different income groups than to just plot the top-ten emitters or countries with claims higher than a certain number.

Table 1: This table seems far too big and busy – I can understand wanting to convey all results for readers, but they run the risk of getting lost in the table and details. I would recommend trying to trim down the number of countries you list here, tied to the countries you discuss at more length in the text. Colors to divide into income groups – what is the rationale for the divisions? Instead of colors, probably best to try and separate them with a row of merged cells, indicating e.g. “High income”, “upper middle” etc. to avoid readability issues if/when printing. Explain the rationale for the division more specifically in the table caption text.

Our response

Thanks for the excellent suggestion to indicate “high-income countries” in Table 1. We have changed accordingly and removed colours. The table provides empirical data for a vast number of countries representing about 90% of global territorial emissions. The value would be reduced if the table had fewer countries. However, we have reduced the number of columns from seven to five.

Figure 2: I quite like this figure and think it is helpful to convey your results and get an idea of where all the countries stand etc. It may be helpful to emphasize the zero lines, and perhaps label the

quadrants / areas indicating as you say in the paper groups of high-income countries where their accountability is higher than their planned emissions, and conversely the larger emissions than accountabilities (so different ways to fulfill their obligations) – the reader could already spot that from your chart. The colors could be more helpful (yellow is hard to read on my screen at least). Would it help your scaling issue (e.g. China) if you put everything in per capita terms? This may then make the comparison of claims and debt impossible though, given the different numbers, so maybe not. While the exploded view (left panel) is useful, it seems a shame to waste so much space on the right panel, as its quite hard to read the country names. But, I can understand the necessity, and the way the exploded view is done is quite helpful for the reader.

Our response

Thank you for these suggestions. We agree that showing these results per capita is indeed more valuable for the reader. We have redone the figure based on the average population during the respective periods for the carbon debt and the excessive carbon claims, i.e., 1990-2022 and 2023-2070. A graph with absolute levels is also provided in Supplementary Figure 1. Lines have been added to emphasize the zero lines and the color palette has been changed to the batlow-palette that optimizes readability and avoids colors that are problematic for color blind people (<https://www.nature.com/articles/s41467-020-19160-7>).

When it comes to labeling the quadrants, we have decided not to do it since the interpretation for the additional carbon accountability is tricky when the data is shown in average per capita terms. Hence, this graph should be used for illustrative purposes since the method uses cumulative population. Nevertheless, illustrating a low or high carbon debt per capita and excessive carbon claim per capita (as given by the axes) does suggest how the carbon debt and excessive carbon claims influence the additional carbon accountability.

Table 2: Needs a more descriptive table caption (at least a full sentence) but I wondered as to the value of this table when reading the paper. It is only referred to once or twice, in two sentences discussing G7 and BRICS, but the disaggregated results do not seem so interesting – I as reader can check in the main table if I want the details. Or if there’s further importance of this breakdown, it needs more elaboration in the table caption and text.

Our response

In Table 3 (former Table 2) we only show G7 and BRICS+ countries to illustrate geopolitical challenges. Here we added a column (Carbon claims per capita 2023-2070) and show that some developing countries within BRICS+ make very large carbon claims. We have extended the analysis because the results suggest that NDCs and NZTs do not perpetuate inequalities like grandfathering:

“On the contrary, many large developed countries take the lead in mitigation, hence respecting Articles 4.4 of the Paris Agreement. Indeed, we found that three middle-income countries in the BRICS+ group – Iran, Russia and China – have larger carbon claims (planned emissions) per capita than any of the G7 countries (Table 3; Supplementary Table 1).” (Lines 232-236)

Analytical approach

Strength of analytical approach – I believe my comments above have generally covered this area as well.

Suggested improvements

Again, see my above comments, but two main improvements:

1. Strengthen the discussion of fairness, and make an argument that the proposed indicator is indeed fair (as per discussion in Validity above)

Our response

See above, we no longer claim that the indicator is fair per se. We have changed the text to:

“The ambition is to build on the Paris Agreement as the internationally agreed climate institution and improve the fairness of its implementation by acknowledging common but differentiated responsibilities for past and future emissions.”

2. Provide a means of comparison for the new indicator – place it in related / relevant literature / estimates. It is hard to judge the impact of the indicator and this study without some reference for reader to compare with.

Our response

See above, we have made extensive sensitivity analysis under the sub-heading “Robustness of the Additional Carbon Accountability” and in Supplementary Figures and Notes.

Clarity and context

See above – the study, while referencing previous work that provides overviews as to fairness considerations, and the need to address normative aspects of carbon budgets explicitly, does not do so to a large degree, and the work could be improved by adding this additional context (as well as the context in terms of how the results compare).

Our response

Under the subheading “Robustness of the Additional Carbon Accountability” we have run our models using other fairness principles than the default equal cumulative per capita.

Also see Supplementary Note 1 for details on the methodological choices tested and Supplementary Figures 6, 7 and 8 for the results.

REVIEWER COMMENTS

Reviewer #1 (Remarks to the Author):

The authors have addressed many of my concerns and made substantial improvements to the manuscript.

Unfortunately in my first reading I did not assess the use of the emissions concept. I understand now that the authors are using territorial emissions rather than consumption-based emissions. This presents a bit of a problem, because the literature on climate justice overwhelmingly supports the use of consumption-based emissions. I understand that consumption-based emissions cannot easily be used for the future period in this study, as the analysis is based on NDCs which (if I am not mistaken) deal with territorial emissions. However, there is no reason that consumption-based emissions could not be used for the historical period, as this data is readily available. I strongly recommend that the authors should use this approach in the main analysis. If not, then at at minimum such an approach should be deployed as part of the sensitivity analysis and results should be discussed in the main text.

The price has been adjusted to \$150, which I think is better and quite reasonable, but the rationale for this needs further clarity. The reference to Fuss et al is fine, but if the range is 100-200 USD/ton for BECCS and 100-300 USD/ton for DACCS, if the distribution is equal it would seem that the correct average is $(150+200)/2 = 175$. Unless Fuss et al indicate that the prices are distributed more toward the bottom of the range? Or unless you are assuming more BECCS than DACCS? Please specify briefly.

One significant issue remains. There is now a substantial literature indicating the limits of NETs. For instance, scaling up BECCS means pressures on land use, water use and biodiversity (particularly in the global South where most of the necessary land would be obtained), which may run against other social and ecological objectives. As for DACCS, it is very energy intensive and therefore deployment at scale would substantially increase energy demand and make decarbonization of the energy system more difficult to achieve. If we account more fully for the limitations faced by NETs, the implication is that "climate debtors" need to scale down emissions faster than they are presently planning with their NDCs. If these limitations cannot be included in the analysis they should at least be discussed.

Reviewer #2 (Remarks to the Author):

Thanks for your answers to my previous comments. In the previous review I mentioned one issue which in my opinion hasn't been addressed sufficiently. I wrote: "Why is the 'Fair National Carbon Accountability Indicator' discounted in proportion to the fair budget that is not used up by some countries. This means that countries with a larger gap between targets and the fair budget, get a higher discount, which seems flawed." I still see this as the main issue with your indicator. In the definition of your indicator, the total excessive claims are reduced for all countries with positive total excessive claims

by multiplication with a factor that is the same for all, thus countries with higher excessive claims receive a higher reduction. Could you explain why you chose to define the indicator that way? An alternative could be to distribute the excessive claims of countries with negative excessive claims to countries with positive excessive claims in proportion to the cumulative population of the country. In that way again the equal cumulative per capita principle would come into play.

Reviewer #3 (Remarks to the Author):

I would like to thank the authors for addressing the concerns I raised in my first review of the work. My major issues having to do with fairness I find have been well addressed, and the what I felt maybe too ambitious claims on fairness have been moderated. I appreciate the detailed responses to my concerns in the responses to reviewers document, and the inclusion of robustness checks etc. which move towards my concern of setting the research in the field / highlighting novelty, along with the addition of the new Robustness section and the additional results / methods in the Supplementary file.

Finally, the figures are overall greatly improved, and easy for the reader to understand (along with the captions).

I think these changes do well to address all my previous concerns.

There were some minor issues I noted throughout the paper:

Line 85: Carbon debt is discussed, but not really defined until Table 1 on line 130; it may be that some readers more broadly interested aren't clear what's meant by carbon debt here.

Line 130 - Table 1: You call it "historic carbon debt" only here, everywhere else its just carbon debt (or carbon debts). An extremely minor issue of consistency.

Line 229: I'm not sure I see a connection between your analysis showing that large developed countries take the lead in mitigation, and grandfathering in NDCs providing legitimacy for status quo levels. Couldn't both be true, your results do indeed show that, but the NDCs also attribute legitimacy to status quo? I suppose it could be that your results do still address the second portion of your statement, in that the risk of perpetuating inequality may not be as high as others would fear.

Line 414, 460, 507, 537 - "Carbondebt" "Excessivecarbonclaims" in the equations without spaces, its quite minor but distracting for the reader

Supplementary information, Line 113: shouldn't "integrate" be "integrates" and line 114 allocation OF not allocation FOR?

Regarding detail in methods section and reproducibility of results:

The methods section does describe the approach in enough detail that one should be able to reproduce the results, as the data are also all supplied by the authors. I did not try and do so from scratch, but

checked how the provided R code corresponded to the methods section.

I can confirm that the code provided does reproduce the results in the paper, and that the data sources etc. all correspond to those specified in the manuscript. I did have some difficulty actually reconciling your methods equations and your R code. I am confident now that the method specified in the text is carried out in your code, but it took a bit of time to read.

E.g. a quick early example from line 43 -> why are you multiplying the variable EmissionsMtCO2 by 44/12?

- I know it is because the data was actually Mt carbon, and you need to multiply it by 3.664 to get to tonnes CO2, but its not clear from the code.

Additionally, as mentioned the equations from methods were difficult to actually identify in the code, but everything is all there, sometimes quite hidden. In the end, there is enough detail for work to be reproduced, and it does not affect the manuscript, so perhaps is not relevant for me as a reviewer to comment on, but it would probably in the future make it easier for other reviewers to verify validity of results if code were clearer or more thoroughly commented when necessary (as code can also be written to be understandable without comments of course).

Thank you for the well-written and interesting paper!

Reviewer #3 (Remarks on code availability):

As I wrote to the authors - I was able to reproduce the results of the paper quite easily, in that I could download their github repository and run the R script using all their provided files. All results seen in the manuscript and supplementary files can be reproduced exactly.

There was not a README file with installation instructions as such, and one would be quite easy to write up for the authors, but it was very straightforward for anyone with an introductory knowledge of R.

As for a usable resource for the community - the code was quite difficult to read in some cases, and perhaps does not lend itself to easy use by third parties.

Answers to Reviewers' Comments

Review Round #2

REVIEWER COMMENTS

Reviewer #1 (Remarks to the Author):

The authors have addressed many of my concerns and made substantial improvements to the manuscript.

1. Unfortunately in my first reading I did not assess the use of the emissions concept. I understand now that the authors are using territorial emissions rather than consumption-based emissions. This presents a bit of a problem, because the literature on climate justice overwhelmingly supports the use of consumption-based emissions. I understand that consumption-based emissions cannot easily be used for the future period in this study, as the analysis is based on NDCs which (if I am not mistaken) deal with territorial emissions. However, there is no reason that consumption-based emissions could not be used for the historical period, as this data is readily available. I strongly recommend that the authors should use this approach in the main analysis. If not, then at minimum such an approach should be deployed as part of the sensitivity analysis and results should be discussed in the main text.

Answer: The Paris Agreement is designed around national policies focused on emissions and removals within each country, hence, the NDCs and net-zero targets concern territorial emissions. Consumption-based emissions are certainly relevant when analysing fairness and could also be relevant to strengthen the implementation of the Paris Agreement by targeting emissions beyond those covered by the territorial scope of the NDCs (see for example <https://www.nature.com/articles/s43247-023-01012-z> for a discussion on how the two perspectives can be combined within the scope of the Paris Agreement). However, given the aim of this study to estimate the Additional Carbon Accountability for countries, defined as each country's responsibility to mitigate or remove CO₂, in addition to achieving its NDC and NZT pledges, we consider it reasonable to use an accounting framework based on what countries have agreed on, i.e. territorial emission reductions. Further, we find it illogical to use different accounting frameworks for past and future emissions. On line 99-101 we clarify that “The ambition is to build on the Paris Agreement, being the internationally agreed climate treaty, and improve the fairness of its implementation by acknowledging common but differentiated responsibilities for past and future emissions.” Given that the aim and ambition of this study concern improvements of the existing political framework and its metric we believe it's reasonable to use territorial emission accounting.

We've added a further clarification on our approach in Methods, on lines 366-370, with references to Hickel 2020 (32), Fanning & Hickel 2023 (33) and Morfeldt et al. 2023 (34):

“Some studies^{32,33} advocate for using consumption-based emission accounting as a basis for fair allocations of the remaining carbon budget. Our analysis uses territorial emission accounting since the Paris Agreement is designed around national policies focused on

emissions and removals within each country and consumption-based emission accounting is not easily incorporated into the Paris Agreement’s transparency framework³⁴.”

Further, we’ve made estimations of the carbon debt based on consumption-based emission accounting that are compared to equivalent estimates using territorial accounting (see Supplementary Figure 9). However, given that the Global Carbon Project only has consumption-based emissions for a subset of the countries analysed in the study and for a smaller number of years (no data is available for 2022 for all countries and full time series are lacking for Norway and Gambia), a sensitivity analysis would need to be designed in a different way from the main analysis which would make comparisons difficult. This comparison is highlighted in the Method, lines 414-417, stating:

“We also provide estimates of countries’ carbon debts using consumption-based emission accounting in comparison to territorial emission accounting (Supplementary Figure 9) based on available data (i.e., for 1990-2021 and excluding Norway and Gambia).”

2. The price has been adjusted to \$150, which I think is better and quite reasonable, but the rationale for this needs further clarity. The reference to Fuss et al is fine, but if the range is 100-200 USD/ton for BECCS and 100-300 USD/ton for DACCS, if the distribution is equal it would seem that the correct average is $(150+200)/2 = 175$. Unless Fuss et al indicate that the prices are distributed more toward the bottom of the range? Or unless you are assuming more BECCS than DACCS? Please specify briefly.

Answer: 150 USD is used to estimate mitigation costs, encompassing both removals and mitigation that could be financed in other countries. If we assumed that the additional accountability would be met by only BECCS and DACCS, in similar proportions, then we could have used this single reference and assumed 175 USD/ton. However, we acknowledge the uncertainty of both scale and cost of CDR and we have added a few references. Still, it is not an aim of the article to analyse the future costs of mitigation and carbon removal, it clearly says: “We assume for simplicity that the average cost for cost-effective measures is 150 USD per ton CO₂ mitigated or removed; this is within the estimated cost range of most CDR methods (see paragraph C.11.1 and Table TS.7 by the by the IPCC).” (line. 525-528). The cost ranges estimated by IPCC AR6 WGIII are larger than in Fuss et al. We also cited another study suggesting lower costs (line 531), this study concludes: “The model results show that DACCS in Europe 2050 could cost between 160 €/tCO₂ and 270 €/tCO₂ with very conservative techno-economic assumptions and between 60 €/tCO₂ and 140 €/tCO₂ using more progressive parameters.” In our manuscript we use 150 USD/ton only for illustrating the magnitude of the costs and compare it with equally uncertain future military spending.

3. One significant issue remains. There is now a substantial literature indicating the limits of NETs. For instance, scaling up BECCS means pressures on land use, water use and biodiversity (particularly in the global South where most of the necessary land would be obtained), which may run against

other social and ecological objectives. As for DACCS, it is very energy intensive and therefore deployment at scale would substantially increase energy demand and make decarbonization of the energy system more difficult to achieve. If we account more fully for the limitations faced by NETs, the implication is that "climate debtors" need to scale down emissions faster than they are presently planning with their NDCs. If these limitations cannot be included in the analysis they should at least be discussed.

Answer: We fully agree with this. We realise that by only referring to BECCS and DACCS in the Methods section the article gives the wrong impression that we believe these technologies will be most important for countries to meet their additional accountabilities. We have therefore added the following after line 533:

"However, since the global potential for BECCS is limited by area and DACCS is limited by scaling and costs⁴¹ (also see Table TS.7 by the IPCC38), a large part of countries' additional carbon accountability may be met by mitigation."

We have also emphasised the importance of additional mitigation (sharper NDCs and NZTs) under Discussion: "Such negative carbon budgets are however not an argument for abandoning national allocations of the carbon budget. On the contrary, it reveals that every ton of future emissions must be accompanied by at least one ton of removal, or additional reductions financed in countries with remaining budgets in the near future, which in turn provides strong incentives for faster mitigation" (lines 319-324).

Reviewer #2 (Remarks to the Author):

4. Thanks for your answers to my previous comments. In the previous review I mentioned one issue which in my opinion hasn't been addressed sufficiently. I wrote: "Why is the 'Fair National Carbon Accountability Indicator' discounted in proportion to the fair budget that is not used up by some countries. This means that countries with a larger gap between targets and the fair budget, get a higher discount, which seems flawed." I still see this as the main issue with your indicator. In the definition of your indicator, the total excessive claims are reduced for all countries with positive total excessive claims by multiplication with a factor that is the same for all, thus countries with higher excessive claims receive a higher reduction. Could you explain why you chose to define the indicator that way? An alternative could be to distribute the excessive claims of countries with negative excessive claims to countries with positive excessive claims in proportion to the cumulative population of the country. In that way again the equal cumulative per capita principle would come into play.

Answer: We agree that this was a weak point in our methodology and would like to thank you for further highlighting it and suggesting an alternative approach. We have decided to incorporate this alternative for redistributing the emission allowances that are not used by countries with negative total excessive carbon claims and we have redistributed this "emission allowance pool" based on equal cumulative per capita access. Hence, amendments have been made to the methodology by redesigning the equation that estimates the Additional Carbon Accountability for each country. Two equations have been added in order to do this: equation 5, which estimates the unused emission

allowances and pools them for redistribution among countries that are assigned an Additional Carbon Accountability, and equation 6, which makes the redistribution based on an equal cumulative per capita approach similar to the one used for the carbon budget. Equation 7 is also adjusted accordingly. The equations are solved iteratively to avoid countries being assigned a negative Additional Carbon Accountability. All results have been updated accordingly along with the sensitivity analysis.

Reviewer #3 (Remarks to the Author):

5. I would like to thank the authors for addressing the concerns I raised in my first review of the work. My major issues having to do with fairness I find have been well addressed, and the what I felt maybe too ambitious claims on fairness have been moderated. I appreciate the detailed responses to my concerns in the responses to reviewers document, and the inclusion of robustness checks etc. which move towards my concern of setting the research in the field / highlighting novelty, along with the addition of the new Robustness section and the additional results / methods in the Supplementary file.

Finally, the figures are overall greatly improved, and easy for the reader to understand (along with the captions).

I think these changes do well to address all my previous concerns.

Answer: Thank you for taking the time to review our manuscript.

6. There were some minor issues I noted throughout the paper:

Line 85: Carbon debt is discussed, but not really defined until Table 1 on line 130; it may be that some readers more broadly interested aren't clear what's meant by carbon debt here.

Answer: We have moved all definitions to Table 1, we think the text benefits from that. On line 78 we explain the concept: "The carbon debt, and with it the responsibility for historical emissions..." and only in Table we specify that we used equal cumulative per capita 1990-2022 for the calculation.

Line 130 - Table 1: You call it "historic carbon debt" only here, everywhere else its just carbon debt (or carbon debts). An extremely minor issue of consistency.

Answer: Thanks for noticing. We have deleted "historic" in the text and in the Table caption.

Line 229: I'm not sure I see a connection between your analysis showing that large developed countries take the lead in mitigation, and grandfathering in NDCs providing legitimacy for status quo levels. Couldn't both be true, your results do indeed show that, but the NDCs also attribute legitimacy to status quo? I suppose it could be that your results do still address the second portion of your statement, in that the risk of perpetuating inequality may not be as high as others would fear.

Answer: We agree that NDCs attribute legitimacy to the present unequal levels of emissions and this is why we allocate additional carbon accountability to high-emitters. However, we find weak support

that NDCs perpetuate inequality since many large developed countries with large carbon debts (e.g. EU) have much more ambitious plans than other developed countries (e.g. Saudi Arabia) and upper middle-income countries. Like grandfathering, NDCs start from the present emission levels, but unlike grandfathering, NDCs do not allocate the global remaining carbon budget to nations based on current emission shares (ref 10 in manuscript). We have changed the text to (lines 190-196):

“Our analysis suggests that the statement above, that the grandfathering characteristic of NDCs and NZTs attributes legitimacy to the present unequal levels of emissions and risks perpetuating inequality¹¹, is only partially true. The timing of countries’ NZTs play a large role in future cumulative emissions. For example, if we assume that a country choose 2070 as NZT over 2050 and that their emissions pathway is linear from 2023 onwards towards the target, then choosing net-zero by 2070 implies 71% higher cumulative emissions compared to choosing 2050 (48/28 years).”

Line 414, 460, 507, 537 - "Carbonebt" "Excessivecarbonclaims" in the equations without spaces, its quite minor but distracting for the reader

Answer: We don't know why spaces were lost in these equations; now it's corrected, and we will make sure to monitor this issue when reviewing the proofs.

Supplementary information, Line 113: shouldn't "integrate" be "integrates" and line 114 allocation OF not allocation FOR?

Answer: Thanks, we have corrected this now.

7. Regarding detail in methods section and reproducibility of results:

The methods section does describe the approach in enough detail that one should be able to reproduce the results, as the data are also all supplied by the authors. I did not try and do so from scratch, but checked how the provided R code corresponded to the methods section.

I can confirm that the code provided does reproduce the results in the paper, and that the data sources etc. all correspond to those specified in the manuscript. I did have some difficulty actually reconciling your methods equations and your R code. I am confident now that the method specified in the text is carried out in your code, but it took a bit of time to read.

E.g. a quick early example from line 43 -> why are you multiplying the variable EmissionsMtCO2 by 44/12?

- I know it is because the data was actually Mt carbon, and you need to multiply it by 3.664 to get to tonnes CO2, but its not clear from the code.

Additionally, as mentioned the equations from methods were difficult to actually identify in the code, but everything is all there, sometimes quite hidden. In the end, there is enough detail for work to be reproduced, and it does not affect the manuscript, so perhaps is not relevant for me as a reviewer to comment on, but it would probably in the future make it easier for other reviewers to verify validity of results if code were clearer or more thoroughly commented when necessary (as code can also be written to be understandable without comments of course).

Answer: Thank you for taking the time to read the R-code. We are grateful for these comments since we want to make our method easily accessible for other users (both researchers and technical experts). Therefore, we decided to include working calculations in both Excel (only main case) and R (with full reproducibility of main case calculations as well as sensitivity analyses). We have gone through the code and added comments to explain what the purpose of each section is and what the code does on individual lines as well as added references to the corresponding equations in the manuscript. We hope that this will make it easier to read and use.

Thank you for the well-written and interesting paper!

Reviewer #3 (Remarks on code availability):

As I wrote to the authors - I was able to reproduce the results of the paper quite easily, in that I could download their github repository and run the R script using all their provided files. All results seen in the manuscript and supplementary files can be reproduced exactly.

There was not a README file with installation instructions as such, and one would be quite easy to write up for the authors, but it was very straightforward for anyone with an introductory knowledge of R.

As for a usable resource for the community - the code was quite difficult to read in some cases, and perhaps does not lend itself to easy use by third parties.

Answer: Thanks again for reviewing the R-code. We have now added a README-file to make it easier for users to install the program, which we hope will make it even more straightforward to make use of the code. In addition, the R-code has been thoroughly commented to assist users' understanding of what it does and what the purpose of different sections is.

REVIEWERS' COMMENTS

Reviewer #1 (Remarks to the Author):

The authors engaged with my concerns even if I believe they did not implement my suggestions in the way I would have liked to see (e.g., they relegated consumption-based accounting to the SI, and stuck with the original price level). These would not be my preferred choices, but at the same time they are not invalid

Reviewer #2 (Remarks to the Author):

I am very happy with the changes now, all my concerns have been addressed. Therefore I would recommend this article for publication. '